# On diffusion posterior sampling via sequential Monte Carlo for zero-shot scaffolding of protein motifs

**James Matthew Young**                                        *matthew.young20@imperial.ac.uk*
*Department of Mathematics*
*Imperial College London*

**O. Deniz Akyildiz**                                         *deniz.akyildiz@imperial.ac.uk*
*Department of Mathematics*
*Imperial College London*

**Reviewed on OpenReview:** *https://openreview.net/forum?id=KXRYY7iwqh*

## Abstract

With the advent of diffusion models, new proteins can be generated at an unprecedented rate. The *motif scaffolding problem* requires steering this generative process to yield proteins with a desirable functional substructure called a motif. While models have been trained to take the motif as conditional input, recent techniques in diffusion posterior sampling can be leveraged as zero-shot alternatives whose approximations can be corrected with sequential Monte Carlo (SMC) algorithms. In this work, we introduce a new set of guidance potentials for describing scaffolding tasks and solve them by adapting SMC-aided diffusion posterior samplers with an unconditional model, Genie, as a prior. In single motif problems, we find that (i) the proposed potentials perform comparably, if not better, than the conventional masking approach, (ii) samplers based on reconstruction guidance outperform their replacement method counterparts, and (iii) measurement tilted proposals and twisted targets improve performance substantially. Furthermore, as a demonstration, we provide solutions to two multi-motif problems by pairing reconstruction guidance with an SE(3)-invariant potential. We also produce designable internally symmetric monomers with a guidance potential for point symmetry constraints. Our code is available at: `https://github.com/matsagad/mres-project`.

## 1 Introduction

Proteins are fundamental to many biological systems. Naturally occurring and defined by an amino acid sequence, they fold to structural conformations that determine their function. Nature, however, has explored but a tiny fraction of the entire protein universe, furthering its reach through evolution at the scale of millions of years. De novo protein design aims to accelerate this process to days or hours. Recent works (Watson et al., 2023; Trippe et al., 2022; Yim et al., 2023; Lin and Alquraishi, 2023) use diffusion models (Sohl-Dickstein et al., 2015) to learn the diverse distribution of protein structures and permit the sampling of new, potentially novel proteins. Consequently, an important design task is imposing the existence of a functional substructure called a motif. While works have demonstrated their ability on this task, the *motif scaffolding problem*, along with its variants, remains a significant challenge.

Amortisation has been the most successful approach, whereby the diffusion model is trained to condition upon the motif as a label (Watson et al., 2023; Lin et al., 2024; Didi et al., 2023). However, with the growing list of design requirements, e.g. secondary structure or binding affinity, training for each may be too expensive, especially when requirements are composed or the underlying model is replaced, prompting the need for retraining. Independent of this, problem-agnostic posterior sampling techniques have been used to solve numerous inverse problems with a diffusion model acting as a prior (Chung et al., 2023; Song et al., 2023). These methods can further guarantee asymptotically exact sampling when paired with sequential Monte

Carlo (SMC) algorithms (Cardoso et al., 2023; Dou and Song, 2023; Wu et al., 2023). By formalising motif scaffolding as an inverse problem, it becomes compatible with posterior samplers and solvable in a zero-shot fashion. In particular, Wu et al. (2023) and Trippe et al. (2022) have done this for the single-motif case by conditioning on a fixed partial view of the protein backbone. However, while they have laid the foundation, their methods are not directly applicable in the event of multiple motifs, which is an important but challenging design task. Moreover, SMC-aided diffusion posterior samplers have not been sufficiently compared in the protein domain, where scaffolding tasks are severely ill-posed and high-dimensional but with solutions that can be quantitatively evaluated. We aim to address these concerns.

In this work, we make the following contributions:

- We propose a set of *guidance potentials* for various scaffolding tasks. Notably, we explore alternatives to masking for guiding the denoising process. These potential functions permit a natural extension to scaffolding multiple motifs while having comparable, if not better, single-motif in-silico performance than masking. Additionally, we provide a potential function for developing monomers with possibly loose internal symmetries that achieves designability in the cyclic and dihedral case.
- We provide a thorough comparison of recent SMC-aided diffusion posterior samplers on motif scaffolding. We identify their design choices, report their empirical costs, and compare them to a baseline in the bootstrap particle filter.
- We demonstrate that it is possible to jointly scaffold multiple motifs in zero-shot by pairing a reconstruction guidance-based SMC sampler with one of our proposed potentials. We also point towards possible improvements in sampler efficiency for future research.

The rest of this paper is organised as follows. In Section 2, we introduce some background to our methods, namely inverse problems, diffusion posterior sampling, and motif scaffolding. We also briefly review relevant literature and existing methods in relation to our algorithms. In Section 3, we present our proposed potentials across several scaffolding tasks while highlighting the intuition behind them. In Section 4, we describe our experimental setup and elucidate our findings. Finally, we highlight limitations and future work in Section 5.

## 2   Background

**Inverse Problems.**   Commonplace in many scientific disciplines, *inverse problems* require estimating a latent quantity $\mathbf{x} \in \mathbb{R}^D$ given its measurement $\mathbf{y} \in \mathbb{R}^d$, an often ill-posed problem. Formally, we define it as

$$\mathbf{y} = \mathcal{A}(\mathbf{x}) + \mathbf{n}, \tag{1}$$

for some possibly nonlinear measurement function $\mathcal{A} : \mathbb{R}^D \to \mathbb{R}^d$ and noise $\mathbf{n} \sim \mathcal{N}(\mathbf{0}, \sigma_v^2 \mathbf{I})$. Eq. (1) analogously defines a likelihood

$$g(\mathbf{y} \mid \mathbf{x}) \propto \exp\left(-\frac{1}{2\sigma_v^2}\|\mathbf{y} - \mathcal{A}(\mathbf{x})\|^2\right),$$

that we generally express in this work as

$$g(\mathbf{y} \mid \mathbf{x}) \propto \exp\left(-\eta \cdot L_{\mathcal{A}}(\mathbf{x}; \mathbf{y})\right),$$

for some $\eta > 0$ and potential $L_{\mathcal{A}} : \mathbb{R}^D \to \mathbb{R}$. We adopt this notation to define arbitrary potentials but remark that our formulations largely remain Gaussian. Conveniently, the composition of several inverse problem constraints can then be represented by summing together individual potentials. Because inverse problems are often underdetermined, we may assume access to a prior distribution on the latent variable $q(\mathbf{x})$ and aim to sample from the posterior distribution $q(\mathbf{x} \mid \mathbf{y})$, which is given under Bayes' rule as $q(\mathbf{x} \mid \mathbf{y}) \propto q(\mathbf{x})g(\mathbf{y} \mid \mathbf{x})$.

**Diffusion Models.**   Diffusion models (Sohl-Dickstein et al., 2015; Song et al., 2020; Ho et al., 2020) are a class of generative models that learn the *time reversal* of a diffusion process applied to a target distribution $q(\mathbf{x}_0)$, the end result of which is a Gaussian $q(\mathbf{x}_T) = \mathcal{N}(\mathbf{0}, \mathbf{I})$. This involves learning the *score* function $\nabla_{\mathbf{x}_t} \log q(\mathbf{x}_t)$, an otherwise intractable quantity, by parameterising it with a neural network. The time reversal forms a bridge between a simple Gaussian and the target distribution, which permits sampling from the target by progressively moving Gaussian samples in the direction of the score. As such, trained diffusion models can be used as priors for complex data distributions and their associated inverse problems.

**Diffusion Posterior Samplers.** Suppose we wish to solve the noisy inverse problem $\mathbf{y} = \mathcal{A}(\mathbf{x}_0) + \mathbf{n}$ with a diffusion (model) prior on $\mathbf{x}_0$. To frame the posterior as our new target, we can modify the reverse process, so it is instead steered by the conditional score $\nabla_{\mathbf{x}_t} \log q(\mathbf{x}_t \mid \mathbf{y})$. This quantity can be decomposed into a sum of two terms using the Bayes' rule as (Chung et al., 2023; Song et al., 2023; Boys et al., 2024)

$$\nabla_{\mathbf{x}_t} \log q(\mathbf{x}_t \mid \mathbf{y}) = \nabla_{\mathbf{x}_t} \log q(\mathbf{x}_t) + \nabla_{\mathbf{x}_t} \log g(\mathbf{y} \mid \mathbf{x}_t), \tag{2}$$

where $\nabla_{\mathbf{x}_t} \log q(\mathbf{x}_t)$ is score of the prior (unconditional model) and $\nabla_{\mathbf{x}_t} \log g(\mathbf{y} \mid \mathbf{x}_t)$ is called the *guidance term*. However, the guidance term is typically intractable, as it involves a direct comparison between a clean measurement and a noisy latent variable. Existing works bridge this gap through approximations that essentially noise the measurement or denoise the latent variable.

Song et al. (2020) define a measurement noising process $\{\mathbf{y}_t\}_{t=0}^T$ that parallels the latent's forward diffusion process, where $\mathbf{y}_0 = \mathbf{y}$ represents the clean measurement and $\nu(\mathbf{y}_t \mid \mathbf{y}_0)$ gives the noisy measurement distribution at time $t$. They subsequently project the latent variables towards or orthogonally onto the constraint manifold $\{\mathbf{x}_t : \mathcal{A}(\mathbf{x}_t) = \mathbf{y}_t\}$ to maintain sample consistency with the measurement throughout the entire denoising process. They make the approximation

$$\nabla_{\mathbf{x}_t} \log q(\mathbf{x}_t \mid \mathbf{y}) = \nabla_{\mathbf{x}_t} \log \int q(\mathbf{x}_t \mid \mathbf{y}_t, \mathbf{y}) \nu(\mathbf{y}_t \mid \mathbf{y}) \mathrm{d}\mathbf{y}_t \approx \nabla_{\mathbf{x}_t} \log \int q(\mathbf{x}_t \mid \mathbf{y}_t) \nu(\mathbf{y}_t \mid \mathbf{y}) \mathrm{d}\mathbf{y}_t$$

$$\approx \nabla_{\mathbf{x}_t} \log q(\mathbf{x}_t \mid \hat{\mathbf{y}}_t) = \nabla_{\mathbf{x}_t} \log q(\mathbf{x}_t) + \nabla_{\mathbf{x}_t} \log g(\hat{\mathbf{y}}_t \mid \mathbf{x}_t), \tag{3}$$

where it is assumed $q(\mathbf{x}_t \mid \mathbf{y}_t, \mathbf{y}) \approx q(\mathbf{x}_t \mid \mathbf{y}_t)$ and $\nu(\mathbf{y}_t \mid \mathbf{y}) \approx \delta_{\hat{\mathbf{y}}_t}$ for $\hat{\mathbf{y}}_t \sim \nu(\mathbf{y}_t \mid \mathbf{y})$. This measurement process is available under a linear structure to the inverse problem (see App. A.2). When $\mathcal{A}(\cdot)$ is a masking transformation, e.g. in image in-painting problems, the sequence $\{\hat{\mathbf{y}}_t\}_{t=1}^T$ is effectively constructed by forward diffusing the measurement $\mathbf{y}$. Here, the guidance term's contribution is equivalent to replacing or interpolating the masked segment of $\mathbf{x}_t$ with $\hat{\mathbf{y}}_t$ (Song et al., 2021). This is known as the *replacement method* (Trippe et al., 2022; Ho et al., 2022). More generally, we will refer to approaches that rely on the score estimate in eq. (3) as replacement methods in our experiments.

The replacement method becomes unsuitable, however, when the measurement process is intractable, as with non-linear inverse problems. Replacement in the linear masking case has also been empirically shown to lose coherence between the masked and unmasked regions (Ho et al., 2022). Alternatively, Chung et al. (2023) propose the diffusion posterior sampling (DPS) method, using the approximation

$$g(\mathbf{y}|\mathbf{x}_t) = \int g(\mathbf{y}|\mathbf{x}_0) q(\mathbf{x}_0|\mathbf{x}_t) \mathrm{d}\mathbf{x}_0 \approx g(\mathbf{y}|\hat{\mathbf{x}}_0(\mathbf{x}_t, t)), \tag{4}$$

where it is assumed $q(\mathbf{x}_0|\mathbf{x}_t) \approx \delta_{\hat{\mathbf{x}}_0(\mathbf{x}_t, t)}$, and $\hat{\mathbf{x}}_0(\mathbf{x}_t, t) = \mathbb{E}[\mathbf{x}_0 \mid \mathbf{x}_t]$ is the reconstructed (clean) sample. This leads to the score estimate

$$\nabla_{\mathbf{x}_t} \log q(\mathbf{x}_t \mid \mathbf{y}) \approx \nabla_{\mathbf{x}_t} \log q(\mathbf{x}_t) + \nabla_{\mathbf{x}_t} \log g(\mathbf{y} \mid \hat{\mathbf{x}}_0(\mathbf{x}_t, t)),$$

where $g(\mathbf{y} \mid \hat{\mathbf{x}}_0(\mathbf{x}_t, t)) = \mathcal{N}(\mathbf{y}; \mathcal{A}(\hat{\mathbf{x}}_0(\mathbf{x}_t, t)), \tilde{\sigma}_t^2 \mathbf{I})$, whose logarithm's gradient is computed via backpropagation. We refer to this method as *reconstruction guidance*.

The above methods rely on single-sample approximations. One can instead devise Gaussian approximations for $q(\mathbf{x}_0|\mathbf{x}_t)$ (Song et al., 2023; Boys et al., 2024), but this too can be a strong assumption. Taking a different approach, one can adopt sequential Monte Carlo (SMC) algorithms (Del Moral et al., 2006) to correct such errors, given that it permits asymptotically exact posterior sampling with added liberties in the algorithm's construction, namely the proposal and intermediate target choices. SMC methods are often used in the context of state space models (SSMs), with their Markovian structure and sequential measurements. A diffusion model's reverse process naturally fits this framework, with the exception of measurements taken at intermediate steps. After all, the inverse problem is only defined on samples in the final target. A workaround is to adopt probabilistic models with a single terminal measurement $\mathbf{y}_0$ or artificially construct a sequence of measurements $\{\mathbf{y}_t\}_{t=1}^T$. However, the former produces low likelihood samples if filtering is only done at the last step, and the latter imposes a realisation of measurements that may not be easily satisfied. We discuss two ways adopted by SMC methods to mitigate these issues.

First, *twisting* (Pitt and Shephard, 1999; Johansen and Doucet, 2008) is an SMC technique that improves sampler efficiency by accounting for measurements in future iterations. It involves tilting the intermediate targets by some twist function $\psi_t$, with $\psi_0 = \psi_T = 1$, maintaining the initial and final targets. Twisting allows the terminal measurement to be accounted for earlier in the process and for smoother transitions between sequential measurements. Second, the reverse diffusion kernel proposal can be further tilted by the likelihood to obtain the (locally) optimal proposal $q_t(\mathbf{x}_t \mid \mathbf{x}_{t+1}, \mathbf{y}_t)$, a choice that minimises the variance of importance weights. This permits the conditional signal to be present beyond just the resampling steps. These building blocks are especially important when working in high dimensions, where a correct but inefficient sampler may face frequent weight degeneracies. Alg. 1 provides a general recipe for SMC diffusion posterior samplers.

The replacement method precisely creates the sequence of measurements that can be used together with SMC. Trippe et al. (2022) apply filtering with the replacement method as part of their SMCDiff algorithm to solve in-painting problems. They maintain the original reverse kernel as their proposal, but replace the masked segment with a noisy version of the measurement. For linear inverse problems in general, Dou and Song (2023) employ the optimal proposal and a noise-sharing technique in their method, FPS-SMC. In the in-painting case, their proposal smoothly interpolates the masked segments with the noisy measurements, which share the same noise applied to their latent counterparts. The consistency of both methods, however, relies on the forward and reverse processes having the same joint distribution, which is often untrue in practice (see (Trippe et al., 2022, App. D.2) and (Dou and Song, 2023, Prop. 4.1)). Maintaining consistency without this assumption, Cardoso et al. (2023) use the same optimal proposal but adopt a terminal measurement model with targets tilted by the likelihood of sequential measurements. Their MCGDiff algorithm solves general linear inverse problems. While they do not explicitly noise the measurements, they use the mean of the measurement distribution as a single sample approximation in eq. (3). In parallel, Wu et al. (2023) use reconstruction guidance together with filtering for solving general inverse problems. They similarly adopt a terminal measurement model but have twist functions and an optimal proposal approximation based on eq. (4). Their method, TDS, provides state-of-the-art performance in motif scaffolding among training-free methods. Specific choices in proposals and twist functions of each sampler, together with their weight updates, are laid out in App. A.2.

Other works that fit within this framework include SVDD (Li et al., 2024), an instance of the bootstrap particle filter (BPF) equipped with the likelihood in eq. (4) which avoids backpropagating through the score model, and PDDS (Phillips et al., 2024), a sampler that reframes the sampling problem as diffusion posterior sampling, leveraging well-behaved diffusion paths, and uses SMC with twisting to correct for errors. A unique feature of these SMC methods is their increasingly accurate posterior sampling at the controlled expense of a larger number of particles.

---

**Algorithm 1:** Recipe for an SMC Diffusion Posterior Sampler

**input** : measurement $\mathbf{y}_0$, proposal $q_t$, likelihood $g_t$,
reverse diffusion transition kernel $p_\theta$,
no. of particles $K$, no. of time steps $T$,
(if replacement method) noisy measurement dist. $\nu$,
(if twisting) twist functions $\psi_t$ (otherwise set $\psi_t = 1$)

**output:** approximate posterior samples $\mathbf{x}_0^{1:K}$

`# Create a sequence of measurements`

Sample $\mathbf{y}_t \sim \begin{cases} \delta_{\mathbf{y}_0} & \text{for reconstruction guidance} \\ \nu(\cdot \mid \mathbf{y}_0) & \text{for replacement method} \end{cases}$,
for $t = 1, \ldots, T$

`# Define transition between targets`

Set $\tilde{f}_{t-1}(\mathbf{z}_{t-1} \mid \mathbf{z}_t)$

$:= \begin{cases} \begin{cases} p_\theta(\mathbf{z}_{t-1} \mid \mathbf{z}_t) & \text{if } t > 1 \\ p_\theta(\mathbf{z}_{t-1} \mid \mathbf{z}_t) g_0(\mathbf{y}_0 \mid \mathbf{z}_{t-1}) & \text{if } t = 1 \end{cases} & \begin{array}{l}\text{for a terminal} \\ \text{measurement}\end{array} \\ p_\theta(\mathbf{z}_{t-1} \mid \mathbf{z}_t) g_{t-1}(\mathbf{y}_{t-1} \mid \mathbf{z}_{t-1}) & \begin{array}{l}\text{for sequential} \\ \text{measurements}\end{array} \end{cases}$

`# Generate samples guided by measurements`

Sample $\bar{\mathbf{x}}_T^{1:K} \sim \mathcal{N}(\mathbf{0}, \mathbf{I})$

Set $\tilde{w}_T^i \leftarrow g_T(\mathbf{y}_T \mid \bar{\mathbf{x}}_T^i)$, for $i = 1, \ldots, K$

**for** $t = T, \ldots, 1$ **do**
  Set $w_t^i \leftarrow \tilde{w}_t^i / \sum_{j=1}^K \tilde{w}_t^j$, for $i = 1, \ldots, K$
  Resample $\mathbf{x}_t^{1:K} \sim \text{Multinomial}(w_t^{1:K}, \ \bar{\mathbf{x}}_t^{1:K})$

  **for** $i = 1, \ldots, K$ **do**
    Sample $\bar{\mathbf{x}}_{t-1}^i \sim q_{t-1}(\cdot \mid \mathbf{x}_t^i)$
    Set $\tilde{w}_{t-1}^i \leftarrow \dfrac{\tilde{f}_{t-1}(\bar{\mathbf{x}}_{t-1}^i \mid \mathbf{x}_t^i)}{q_{t-1}(\bar{\mathbf{x}}_{t-1}^i \mid \mathbf{x}_t^i)} \dfrac{\psi_{t-1}(\bar{\mathbf{x}}_{t-1}^i)}{\psi_t(\mathbf{x}_t^i)}$
  **end**
**end**

---

**Motif Scaffolding.** A protein's function is characterised by several of its structural domains. As a result, motifs from known domains can act as a foundation that, when scaffolded, can give rise to proteins with specific functional or conformational properties. The process of fixing a substructure is non-trivial, however, as the protein's folding dynamics must be respected. Generative models on protein structure can be leveraged, given their inherent understanding of well-folded structures. RFDiffusion (Watson et al., 2023) and Genie2 (Lin et al., 2024) are current state-of-the-art diffusion models for this task, capable of scaffolding several benchmark motifs (Watson et al., 2023), from small molecule binding sites to viral epitopes. More generally, one can also scaffold multiple motifs simultaneously, e.g., to map out different functional domains. *Multi-motif scaffolding* is more challenging as interactions between motifs must be accommodated by the scaffold. This differs from scaffolding a single motif with fixed discontiguous segments, as each motif can be oriented independently.

Another useful task is the generation of oligomers embodying some point symmetry, i.e. through the self-assembly of multiple identical monomers (Watson et al., 2023; Ingraham et al., 2023). We additionally bring attention to the design of internally symmetric monomers which fits with current monomeric generative efforts. *Symmetric motif scaffolding* demands a symmetric protein structure with each asymmetric subunit containing a copy of the motif. For example, the SARS-CoV-2 spike protein, a $C_3$ symmetric trimer, may be inhibited by attaching a similarly symmetric binder—a feat RFDiffusion demonstrated in-silico.

In these scaffolding tasks, it is common to sample from a range of positions in the backbone and fix the motif in place. Wu et al. (2023) explored allowing multiple motif placements to be jointly considered during the denoising process. We refer to this as *scaffolding with degrees of freedom*. All the scaffolding tasks are illustrated in Fig. 1.

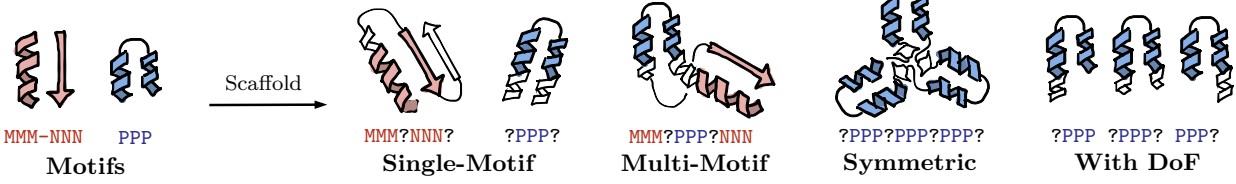

Figure 1: **Different motif scaffolding tasks.** The motif in blue is contiguous, and the other in red is discontiguous. Scaffolds are illustrated in white.

**Related Work.** We focus on solving motif scaffolding problems using an unconditional protein backbone diffusion model without additional training. Several works have previously explored the single-motif case in a similar vein. SMC-based methods include SMCDiff, a replacement method paired with ProtDiff (Trippe et al., 2022), and TDS, a reconstruction guidance method paired with FrameDiff (Yim et al., 2023). A similar work, Chroma (Ingraham et al., 2023), provides a suite of conditioners built around an unconditional model and addresses substructure motifs through reconstruction guidance and a root mean square deviation (RMSD) metric. We believe the latter idea can be incorporated without additional scaling hyperparameters.

On scaffolding multiple motifs, diffusion models have been trained to condition on pairwise residue $C\alpha$ distances, allowing motifs to be represented independently of each other. Genie2, an extension of Genie (Lin and Alquraishi, 2023) under this scheme, achieves in-silico success on some of its curated multi-motif benchmarks. On the other hand, Castro et al. (2024) demonstrate experimental success in jointly scaffolding three epitope motifs using RFJoint2 (Wang et al., 2021), a modified RoseTTAFold network for in-painting. Taking inspiration from these efforts while side-stepping the requirement for training, we explore using distances as part of the inverse problem.

Lastly, on diffusion posterior sampling, replacement methods such as FPS-SMC (Dou and Song (2023)) and MCGDiff (Cardoso et al. (2023)) have not been explored in the motif scaffolding setting. SMC methods in general have also yet to be compared with a baseline such as the bootstrap particle filter (BPF). Formalising the scaffolding inverse problems, we aim to investigate which samplers and formalisations perform best for the tasks.

# 3 Guidance Potentials for Motif Scaffolding

The motif scaffolding problem has structure and sequence requirements, only the former of which is accessible to a typical backbone diffusion model. Hence, our focus is on the structural aspect of the problem. In this section, we provide different ways (in particular a number of potential functions) to encode the scaffolding problems to be solved by diffusion posterior samplers.

For convenience, we work with the flattened representation $\mathbf{x} \in \mathbb{R}^{3N}$, a protein backbone with $N$ residues, each represented by its three-dimensional C$\alpha$ coordinates. From these coordinates, we may represent the protein as a set of rigid body frames, where the $i$th residue is given by $\mathbf{T}_i(\mathbf{x}) = (\mathbf{R}_i(\mathbf{x}), \mathbf{t}_i(\mathbf{x})) \in \text{SE}(3)$, a pair of a rotation matrix and a translation vector with respect to a global reference frame. For a proper rigid transformation $\mathbf{T} \in \text{SE}(3)$, we denote by $\mathbf{T} \circ \mathbf{x}$ the result of applying the transformation to each of the residue coordinates of the protein.

## 3.1 Single-Motif Scaffolding

We define the motif and scaffold index sets $(\mathcal{M}, \mathcal{S})$ as a partition over all the backbone coordinate values, with each residue's coordinates belonging to the same set. Furthermore, we assume an ordering on $\{\mathcal{M}_i\}_{i=0}^{|\mathcal{M}|}$ and $\{\mathcal{S}_i\}_{i=0}^{|\mathcal{S}|}$, first according to residue number and then coordinate axis. Given the C$\alpha$ coordinates of a motif $\mathbf{m} \in \mathbb{R}^{|\mathcal{M}|}$, we set our measurement to be $\mathbf{y} = \mathbf{m}$. The (single-) motif scaffolding problem requires sampling from the distribution $p(\mathbf{x} \mid \mathbf{x}_{\mathcal{M}} = \mathbf{y})$, where the conditioned equality holds when the motif, oriented in whichever way, appears in the backbone region specified by the motif index set, i.e. there exists a $\mathbf{T}^* \in \text{SE}(3)$ where $\mathbf{x}_{\mathcal{M}} = \mathbf{T}^* \circ \mathbf{y}$.

This task is commonly framed as a linear inverse problem by fixing the motif's orientation and conditioning on a partial view of the backbone. Here, we use a linear transformation $\mathcal{A} := \mathbf{A}_{\mathcal{M}} \in \mathbb{R}^{|\mathcal{M}| \times 3N}$, given by $(\mathbf{A}_{\mathcal{M}})_{ij} = \delta_{\mathcal{M}_i, j}$, to mask the backbone and allow a direct comparison with the motif, which is translated to share the same centre-of-mass (CoM). The corresponding potential is thus given by

$$L_{\text{mask}}(\mathbf{x}; \mathbf{y}, \mathcal{M}) = \|\mathbf{y} - \mathbf{A}_{\mathcal{M}}\mathbf{x}\|^2. \tag{5}$$

Throughout, we equivalently use $\mathbf{x}_{\mathcal{M}}$ in place of $\mathbf{A}_{\mathcal{M}}\mathbf{x}$. This approach is taken in several posterior sampling works (Trippe et al., 2022; Wu et al., 2023), drawing parallels to image inpainting. However, unlike images, a protein motif can further be oriented in 3D space, and fixing it disrupts the equivariance of the denoising process while posing questions on how multiple motifs may be handled. To address these concerns, we now introduce guidance potentials we develop in this work as alternatives to the masking potential (5).

**Distance.** Notably, Wu et al. (2023) improved masking performance by parameterising over a finite set of possible motif orientations. This is likely because we impose a narrower path towards the posterior distribution without the additional degree of freedom. Motivated by this, we consider conditioning on the full pairwise C$\alpha$ distances of the unmasked backbone. Denote the pairwise Euclidean distance matrix derived from two flattened 3D coordinate vectors $\mathbf{u}, \mathbf{v} \in \mathbb{R}^{3n}$ as $\mathbf{D}_{\mathbf{u}, \mathbf{v}}^{\text{euc}} \in \mathbb{R}^{n \times n}$, where $(\mathbf{D}_{\mathbf{u}, \mathbf{v}}^{\text{euc}})_{ij} = \|\mathbf{t}_i(\mathbf{u}) - \mathbf{t}_j(\mathbf{v})\|$. We define the potential as

$$L_{\text{dist}}(\mathbf{x}; \mathbf{y}, \mathcal{M}) = \|\mathbf{D}_{\mathbf{y}, \mathbf{y}}^{\text{euc}} - \mathbf{D}_{\mathbf{x}_{\mathcal{M}}, \mathbf{x}_{\mathcal{M}}}^{\text{euc}}\|_F^2 \tag{6}$$

where $\|\cdot\|_F$ is the Frobenius norm. Contrary to masking, this approach yields an orientation-free motif representation. Its indifference to reflections, however, violates the chirality of proteins, and we explore this drawback in our experiments. The above is akin to an unnormalised dRMSD metric (AlQuraishi, 2019).

**Frame-Distance.** We propose to break the reflection-invariance of the distance potential (6) by further conditioning on the backbone's pairwise orientation deviations within its frame representation. Similarly, we define distances between the rotation matrices $\mathbf{R}_i(\mathbf{x})$. First, we remove the dependence on the protein's current orientation by computing the relative rotations $\mathbf{R}_j(\mathbf{x})^\top \mathbf{R}_k(\mathbf{x})$ for every residue pair $(j, k)$. Then, we use the result measuring the cosine of the angular difference between two rotation matrices

$$d_{\cos}(\mathbf{R}_1, \mathbf{R}_2) = \frac{1}{2}(\text{Tr}\left(\mathbf{R}_1 \mathbf{R}_2^\top\right) - 1),$$

to compare the sample's relative rotations with those of the motif. We define

$$L_{\text{chiral}}(\mathbf{x}; \mathbf{y}, \mathcal{M}) = \|\mathbf{1}\mathbf{1}^\top - \mathbf{D}^{\cos}_{\mathbf{y}, \mathbf{x}_{\mathcal{M}}}\|_F^2,$$

$$\text{where } (\mathbf{D}^{\cos}_{\mathbf{y}, \mathbf{x}_{\mathcal{M}}})_{ij} = d_{\cos}(\mathbf{R}_i(\mathbf{y})^\top \mathbf{R}_j(\mathbf{y}), \ \mathbf{R}_i(\mathbf{x}_{\mathcal{M}})^\top \mathbf{R}_j(\mathbf{x}_{\mathcal{M}})).$$

We keep the angle in its cosine form due to the instability of arccosine gradients near one. Summing with the previous formulation, we have the potential

$$L_{\text{framedist}}(\mathbf{x}; \mathbf{y}, \mathcal{M}) = L_{\text{dist}}(\mathbf{x}; \mathbf{y}, \mathcal{M}) + \eta_{\text{chiral}} \cdot L_{\text{chiral}}(\mathbf{x}; \mathbf{y}, \mathcal{M}), \tag{7}$$

where $\eta_{\text{chiral}}$ is a hyperparameter to scale the magnitude of the chiral contribution.

**Frame-Aligned Point Error (FAPE).** A similar idea of an SE(3)-invariant measure with chiral properties is the frame-aligned point error (FAPE)—the main component of the AlphaFold2 loss function (Jumper et al., 2021). FAPE defines a distance metric between two sets of rigid body frames of the same dimension and attains a value of zero when the two sets of frames are identical up to proper rigid transformations. To match the magnitude of the masking approach, we modify the original formulation to yield the potential

$$L_{\text{fape}}(\mathbf{x}; \mathbf{y}, \mathcal{M}) = \frac{1}{|\mathcal{M}|} \sum_{i=1}^{|\mathcal{M}|} L_{\text{mask}}\left(\mathbf{T}_i(\mathbf{x}_{\mathcal{M}})^{-1} \circ \mathbf{x}; \ \mathbf{T}_i(\mathbf{y})^{-1} \circ \mathbf{y}, \mathcal{M}\right),$$

where $\mathbf{T}^{-1}$ is the inverse of the transformation $\mathbf{T}$. For each motif residue, we essentially set its C$\alpha$ atom as the global reference, allowing both the specified backbone region and the motif to coincide at the residue and their deviation to be computed.

**Root Mean Squared Deviation (RMSD).** A common measure for structural similarity between protein structures is through their root mean squared deviation (RMSD). For protein backbones $\mathbf{u}, \mathbf{v} \in \mathbb{R}^{3n}$, each with $n$ residues, we have

$$\text{RMSD}(\mathbf{u}, \mathbf{v}) = \min_{\mathbf{T}^* \in \text{SE}(3)} \frac{1}{\sqrt{n}} \|\mathbf{u} - \mathbf{T}^* \circ \mathbf{v}\|.$$

Similarly, we define a potential matching the form and magnitude of masking

$$L_{\text{rmsd}}(\mathbf{x}; \mathbf{y}, \mathcal{M}) = |\mathcal{M}| \cdot \text{RMSD}(\mathbf{x}_{\mathcal{M}}, \mathbf{y})^2 = \min_{\mathbf{T}^* \in \text{SE}(3)} L_{\text{mask}}(\mathbf{x}; \mathbf{T}^* \circ \mathbf{y}, \mathcal{M}).$$

Here, we can obtain the minimiser transformation $\mathbf{T}^* = (\mathbf{R}^*, \mathbf{t}^*)$ using the Kabsch algorithm (Kabsch, 1976) or directly compute the RMSD through quaternions (Coutsias et al., 2004). Both methods are differentiable.

A consequence of the proposed potentials' SE(3)-invariance is their gradients, used in reconstruction guidance, maintain the equivariance of the denoising process—i.e. the global orientation of noisy samples do not affect how they are denoised.

## 3.2 Multi-Motif Scaffolding

Now, suppose the index sets $\{\mathcal{M}^1, \ldots, \mathcal{M}^M, \mathcal{S}\}$ form a partition over the backbone coordinates and are each ordered according to residue number and coordinate axis. Given motifs $\mathbf{m}_1, \ldots, \mathbf{m}_M$, we define the measurement $\mathbf{y} \in \mathbb{R}^{3N}$ as $\mathbf{y}_{\mathcal{S}} = \mathbf{0}$ and $\mathbf{y}_{\mathcal{M}^i} = \mathbf{m}_i$ for $i \in [1, M]$. The multi-motif scaffolding problem requires sampling from the distribution $p(\mathbf{x} \mid \mathbf{x}_{\mathcal{M}^1} = \mathbf{y}_{\mathcal{M}^1}, \ldots, \mathbf{x}_{\mathcal{M}^M} = \mathbf{y}_{\mathcal{M}^M})$, with the conditioned equalities similar as before, i.e. there exists $\mathbf{T}_i^* \in \text{SE}(3)$ where $\mathbf{x}_{\mathcal{M}^i} = \mathbf{T}_i^* \circ \mathbf{y}_{\mathcal{M}^i}$ for $i \in [1, M]$.

Unlike in the single motif case, the masking approach fixes the motif-to-motif orientations and severely underrepresents the posterior distribution. Instead, we may use the other proposed approaches to keep each motif's orientation independent of others. This can be achieved by summing the potentials

$$L^{\mathcal{M}^{1:M}}\left(\mathbf{x}; \mathbf{y}, \mathcal{M}^1, \ldots, \mathcal{M}^M\right) = \sum_{i=1}^{M} L^{\mathcal{M}^i}\left(\mathbf{x}; \mathbf{y}_{\mathcal{M}^i}, \mathcal{M}^i\right),$$

where $L^{\mathcal{M}^i}$ is the chosen potential corresponding to the $i$th motif. While it is possible to use different potentials for each motif, we choose to keep it fixed in our analyses.

### 3.3 Symmetric Generation

For some point symmetry group in $\mathbb{R}^3$, define $\mathcal{G} = \{\mathbf{g}_k\}_{k=0}^{n-1}$ as the set of all its symmetry operations. We consider designing internally symmetric monomers as we focus on diffusion models that produce a single chain. However, our formulation can analogously be applied to models supporting multiple chains to design symmetric oligomers by treating each subunit as a monomer. Suppose the chain is composed of $n$ identical subunits with $N$ divisible by $n$. In dealing with 3D atom coordinates, $\mathcal{G}$ is a set of transformation matrices in $\mathbb{R}^{3 \times 3}$. Without loss of generality, we select $\mathbf{g}_0$ as the identity matrix. We can then construct the potential

$$L_{\mathcal{G}}(\mathbf{x}) = \|(\mathbf{A}_{\mathcal{G}} - \mathbf{I}_{3N})\mathbf{x}\|^2,$$

where our measurement is implicitly set to $\mathbf{y} = \mathbf{0}$ and $\mathbf{A}_{\mathcal{G}} \in \mathbb{R}^{3N \times 3N}$ is given by

$$\mathbf{A}_{\mathcal{G}} = \left[ \begin{array}{c|c} \begin{array}{c} \mathrm{diag}(\mathbf{g}_0, \ldots, \mathbf{g}_0) \\ \vdots \\ \mathrm{diag}(\mathbf{g}_{n-1}, \ldots, \mathbf{g}_{n-1}) \end{array} & 0 \end{array} \right],$$

composed of block diagonals $\mathrm{diag}(\mathbf{g}_k, \ldots, \mathbf{g}_k) \in \mathbb{R}^{3N/n \times 3N/n}$. Effectively, this constrains the generated protein to be identical to several symmetric projections of its first subunit. This partitions the chain into $n$ contiguous segments representing each subunit. However, one can shuffle the block diagonals between group operations to render the subunits discontiguous. We demonstrate this process for cyclic and dihedral symmetries.

#### 3.3.1 Cyclic and Dihedral Symmetries

Proteins with cyclic symmetry $C_n$ are invariant to any integer multiple rotations of $2\pi/n$ with respect to a given axis. Without loss of generality, we choose to work with the $z$-axis and accordingly translate $\mathbf{x}$ so its CoM lies on it. Denote $\mathbf{R}_{a,\theta} \in \mathbb{R}^{3 \times 3}$ as the rotation matrix that rotates a vector anti-clockwise about the $a$-axis by an angle of $\theta$. As such, we have the set of rotations $\mathcal{G}_{C_n} = \{\mathbf{R}_{z,2\pi k/n}\}_{k=0}^{n-1}$. While any ordering of the set $\mathcal{G}$ is conducive to producing a cyclic protein, it may be favourable to have adjacent angles for adjacent sub-sequences, e.g. $\mathbf{g}_k = \mathbf{R}_{z,2\pi k/n}$.

Proteins with dihedral symmetry $D_n$ similarly have $C_n$ symmetry in one axis but have $C_2$ symmetry in another axis orthogonal to the first. We choose the $z$- and $y$-axes as primary and secondary axes of symmetry and translate $\mathbf{x}$ to have its CoM at the origin. Thus, we have $\mathcal{G}_{D_n} = \mathcal{G}_{C_n} \cup \{\mathbf{R}_{z,2\pi k/n}\mathbf{R}_{y,2\pi}\}_{k=0}^{n-1}$. We may then substitute $\mathbf{A}_{\mathcal{G}_{C_n}}$ and $\mathbf{A}_{\mathcal{G}_{D_n}}$ into our symmetry potential to impose cyclic and dihedral symmetries.

#### 3.3.2 Symmetric Motif Scaffolding

In addition to symmetric constraints, we may also condition the existence of a motif. Note that we can assume the motif is local to exactly one subunit and $n - 1$ copies exist in the others. Otherwise, if the motif lies on the boundary between subunits, we can redefine the motif to be the residues entirely situated in one subunit. Suppose then that the motif is in the first subunit, i.e. $\mathcal{M}_i \leq 3N/n$ for all $i$.

We may adapt the masking approach with a motif measurement $\mathbf{y} = \mathbf{m}$ and define the potential

$$L_{\mathcal{G},mask}(\mathbf{x}; \mathbf{y}, \mathcal{M}) = \left\| \begin{bmatrix} \mathbf{e}_{\mathcal{M}_1} & \ldots & \mathbf{e}_{\mathcal{M}_{|\mathcal{M}|}} \end{bmatrix} \mathbf{y} - (\mathbf{A}_{\mathcal{G}} - \mathbf{I}_{3N} + \mathrm{diag}(\mathbb{1}_{\mathcal{M}})) \mathbf{x} \right\|^2$$

where $\mathbf{e}_i \in \mathbb{R}^N$ is the $i$th standard basis vector. The additional diagonal term unmasks the motif indices and asserts it is equal to the chosen motif $\mathbf{y}$. We remark that this preserves the linearity of the inverse problem. More generally, e.g. for the distance approaches, we can simply add the symmetry potential $L_{\mathcal{G}}$ onto the motif potential. For masking, however, we choose to combine the linear measurement functions so a single matrix may be supplied for supporting posterior samplers.

# 4 Results

## 4.1 Experimental Setup

Similar to existing works (Trippe et al., 2022; Watson et al., 2023; Lin and Alquraishi, 2023), we use an in silico *self-consistency pipeline* for measuring the quality of protein backbones. The procedure involves an inverse-folding network and a structure prediction network. We use ProteinMPNN (Dauparas et al., 2022) and ESMFold (Lin et al., 2022), respectively. First, the inverse-folding network predicts each generated backbone's representative amino-acid sequences. Then, the structure prediction network folds these sequences into structures. The *self-consistency root mean squared deviation (scRMSD)* between the generated and predicted backbones are then computed, and the smallest is reported. The premise of this technique is that generated backbones possessing natural structures are likely to be represented consistently across orthogonal methods. We use the available in-silico design pipeline provided by the authors of Genie2[1]. The set-up is summarised in Fig. 2.

**Designability and Diversity Metrics.** We adopt a similar designability criterion as Lin et al. (2024). We consider a protein backbone as *designable* if it deviates with the most similar predicted design by at most two Angstroms (scRMSD $\leq$ 2A) and if the designs are confidently predicted (pLDDT $\geq$ 70). For motif scaffolding tasks, we consider a scaffold to be successful if it is designable as above, the motif is present in the predicted backbone within one Angstrom in alignment deviation (motif RMSD $\leq$ 1A), and there is a low predicted alignment error (pAE) between residues (pAE $\leq$ 5). For multi-motif scaffolding, every motif must be within one Angstrom. We remark that scRMSD in motif scaffolding differs from the unconditional setting, as inverse-folded sequences are conditioned to fix the motif's sequence. Furthermore, as success rates can be misleading with identically designed structures, we also measure sample *diversity*. Designs are grouped using single-linkage hierarchical clustering with a distance threshold given by a TM-score of 0.6. We report the number of unique successful scaffolds for motif scaffolding tasks.

**Conditional Setup.** In our analyses, we use Genie (Lin and Alquraishi, 2023), an unconditional protein backbone diffusion model. Specifically, we use GENIE-SCOPE-128 and GENIE-SCOPE-256, models trained on proteins from the SCOPe dataset (Fox et al., 2014), capable of generating proteins of up to 128 and 256 residues, respectively. For single-motif problems, where the overall length is at most 128 residues, we use GENIE-SCOPE-128 for quicker inference times. For the multi-motif case, with problems requiring longer samples, we use GENIE-SCOPE-256. In each model, samples are denoised for $T = 1000$ steps. We parallelise our setup across two NVIDIA GeForce RTX-3090 GPUs to accelerate the process but limit each motif problem to 32 backbones per posterior sampler.

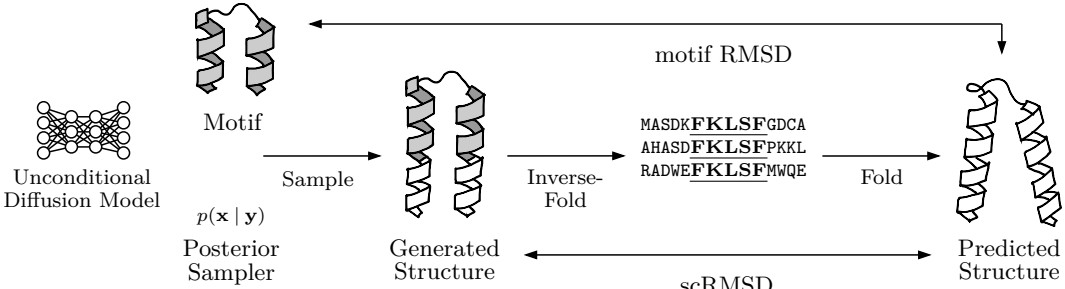

Figure 2: **An overview of the motif scaffolding experimental setup.** Protein backbones are first sampled from the conditional setup with the motif as an observation. These generated structures are then inverse-folded with the motif sequence fixed and folded back into structures. Finally, metrics such as self-consistency RMSD and motif RMSD are computed between the predicted structure and both the generated structure and the motif.

---

[1]The repository is available at `https://github.com/aqlaboratory/insilico_design_pipeline`.

We examine the performance of posterior samplers across different motif scaffolding benchmarks. Under the replacement method, we use FPS-SMC (FPSSMC) and the noiseless version of MCGDiff (MCGDIFF) and compare them with the bootstrap particle filter (BPF) as a baseline. In particular, we define the BPF likelihoods by constructing a sequence of measurements $\{\mathbf{y}_t\}_{t=1}^T$ either via forward noising the motif as in SMCDiff (BPF-FW) or using the reverse process as in FPS (BPF-BW). These are limited to linear inverse problems and thereby use the masking approach. Under reconstruction guidance, we use TDS paired with masking (TDS-MASK), distance (TDS-DIST), frame-distance (TDS-FRAME-DIST), FAPE (TDS-FAPE), and RMSD (TDS-RMSD) guidance potentials.

We set the potential scales to be $\eta_t = 1/2\sigma_v^2\bar{\alpha}_t$ for replacement methods and $\eta_t = 1/((1 - \bar{\alpha}_t) + \sigma_v^2)$ for reconstruction guidance, under a fixed likelihood standard deviation $\sigma_v = 0.05$. We also fix the number of particles to be $K = 8$ and perform adaptive (residual) resampling, with resampling taking place when the effective sample size becomes less than half the particles. We additionally perform hyperparameter searches (see App. D.1) jointly for each sampler and corresponding guidance potential based on two motif problems: 3IXT and 1PRW—a contiguous and discontiguous problem, respectively.

## 4.2 Single-Motif Scaffolding

We sampled proteins conditioned on motifs from the RFDiffusion benchmark (Watson et al., 2023). Twelve of the 24 problems had at least one successful solution among the 32 designs generated for each problem. Fig. 3 summarises the performance of samplers across the benchmarks. Among the methods, TDS-RMSD solved the most, with ten problems. The non-linear potentials paired with TDS showed a comparable, if not higher, number of unique successes over TDS-MASK for several problems. On the one hand, this shows the viability of masking for single-motif scaffolding, maintaining the linearity of the inverse problem and being broadly applicable to many posterior samplers. On the other hand, the non-linear potentials, with their comparable performance and generalisability to multiple motifs, present a case for being a drop-in replacement.

We tested our BPF samplers as a baseline to see if resampling provides a sufficient conditional signal to warrant keeping the proposal unchanged. Results of the replacement-based BPFs and TDS hyperparameter search (App. D.1), in which a zero guidance scale yields a twisted reconstruction guidance-based BPF, indicate that tilting the proposal substantially improves performance and may even be necessary for complex motifs. Intuitively, when weights are degenerate and all particles are identical, BPF chooses the most favourable noise leading to the motif's formation. But as motifs grow in size or deviate from common substructures, even a crude approximation of the conditional score is crucial to move in the right direction in such a high-dimensional space.

Between the replacement methods, we found MCGDIFF to be overall more successful than FPSSMC. The asymmetry between Genie's forward and reverse processes was likely incompatible with the latter's consistency requirements. However, both methods were outperformed by TDS paired with various guidance potentials. This comes despite reconstruction guidance viewing the motif's formation as an optimisation problem, uncertain to be solved at the end of sampling, unlike in replacement methods.

Given the diversity of motifs, we provide additional context to elucidate our results. Motif problems 1BCF, 1PRW, and 2KL8 are discontiguous, restricting the possible conformations the scaffold can take, and have scaffolds substantially shorter than the motif. Because of this, valid scaffolds have minimal variation, resulting in low diversity among successful designs. Hence, we are more interested in whether success is achieved rather than its frequency. State-of-the-art methods yield one unique success for each of the three problems, albeit with the low diversity due to one thousand backbones being sampled (Lin et al., 2024). In 32 samples, TDS-MASK, TDS-FAPE, and TDS-RMSD were successful in all three. The distance approaches had mixed successes on this front but had several designs that missed out on only one of the four success criteria. In contrast, more successes were achieved for contiguous motif problems 1YCR, 3IXT, and 6EXZ, with TDS-FRAME-DIST and TDS-RMSD attaining up to six unique successes in the 6EXZ problems.

Notably, problems such as 1QJG, 5TRV, and 7MRX had several designs with low scRMSD but few with low motif RMSD. Consider, for example, the 1QJG motif made up of three residues separated by some scaffolding. Many protein backbones can satisfy three residues being at these distances away from each other, but only a subset of those backbones can be achieved by folding a protein with a particular trio of amino acids in its

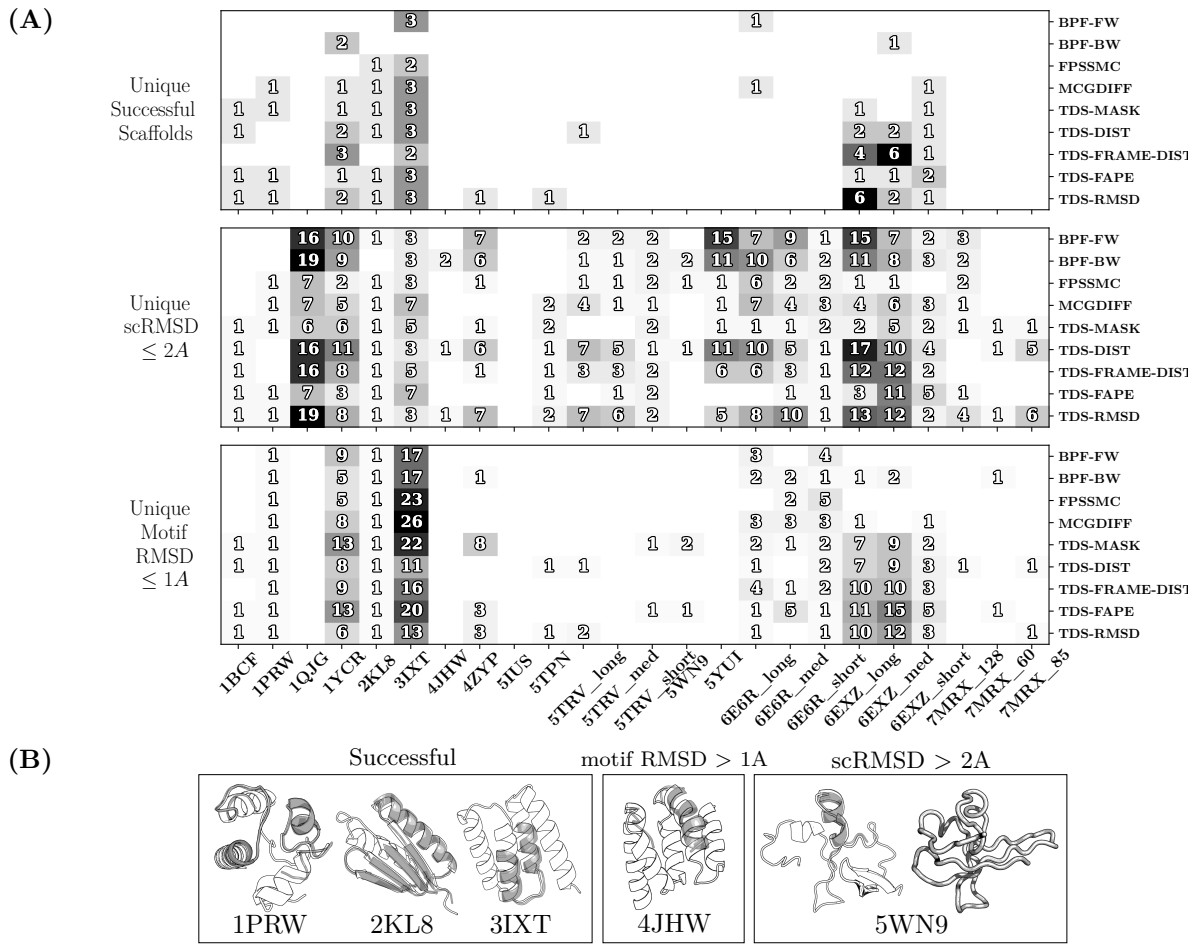

Figure 3: **(A) Performance of sampling methods on the 24 motif scaffolding benchmarks.** Thirty-two backbones are sampled from each method across all the motif problems. Scaffolds that are successful and those which meet at least one of the main success criteria are reported according to their unique count. **(B) Examples of the designed scaffolds.** The motif, in grey, is aligned with the scaffold, in white. Most unsuccessful scaffolds either do not possess the motif in full or have poor self-consistency.

sequence. In other words, the motif's sequence is critical to defining the support of our target distribution, which is currently underspecified. We believe that our setup is already successful in sampling from the much larger support of natural structures possessing the backbone folds of the motif. However, more samples are required to cover the subset of structures with the motif's sequence. To test this hypothesis, we modify the self-consistency pipeline to remove the conditioning of the motif sequence in App. D.2. Indeed, our setup had more successes overall and had solutions for all seven of the 1QJG, 5TRV, and 7MRX problems, indicating that the backbones generated could be designed with an alternate sequence but were incompatible with the motif sequence. Given our setup only conditions on structure, we believe incorporating sequence information can substantially improve performance in select motifs.

We remark that limiting our setup to a single motif placement in each problem may have resulted in too restrictive motif placements. Common approaches that could be employed are uniformly sampling valid placements or allowing multiple placements to be considered during denoising, as done by Wu et al. (2023). We exclude these in the study to minimise the variance of our experiments, given our choice of relatively smaller sample sizes. With more samples, the diversity of successes is expected to diminish. It may then be preferable to use methods that generally produce more diverse samples, as they have a higher ceiling for unique successes. We hypothesise that methods that do not fix the motif's orientation allow for more flexibility during denoising, enabling designs to be more diverse.

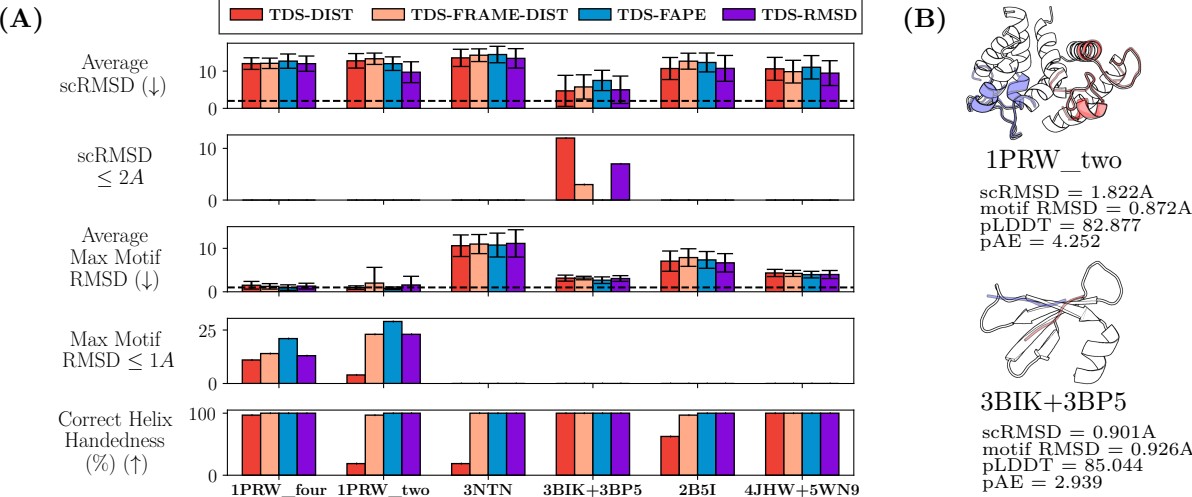

Figure 4: **(A) Success metrics of sampling methods on the six multi-motif scaffolding benchmarks.** Thirty-two scaffolds are sampled for each problem. Values for a pass in each criterion are denoted by the dashed line. Error bars shown are one standard deviation from the mean. Only samples with correct handedness were considered. **(B) Examples of successful designs from the 512 samples generated via TDS-rmsd.** The motifs, in colour, are aligned with the scaffold, in white.

### 4.3  Multi-Motif Scaffolding

We tested the non-linear guidance potentials paired with TDS on the six multi-motif problems from the Genie2 benchmark. In our initial assessment, where 32 designs were produced for each problem, none of the methods succeeded, but some designs narrowly missed the success criteria. Our reference, conditionally trained model Genie2, solved four problems with less than 20 unique successes among 1000 samples for each problem. To match the same order of magnitude in samples, we chose the most promising method, TDS-rmsd, and produced 512 samples for problems 1PRW_two and 3BIK+3BP5. Our choice is motivated by the observation that some TDS-rmsd designs in the two problems were successful when an alternate motif sequence was used. In our larger batch of samples, TDS-rmsd found one unique success in each problem. Fig. 4 summarises our findings.

First, we review our initial assessment, where there is an apparent dynamic between the distance approaches. We found that, independent of the number of motifs, TDS-dist was more likely to produce reflections when motifs had discontiguities, as indicated by its lower proportion of samples with correct handedness. TDS-frame-dist had fixed this issue in virtually all samples, only behind TDS-fape and TDS-rmsd, which maintained the correct handedness throughout. Unlike the latter two, TDS-frame-dist has an additional hyperparameter $\eta_{\text{chiral}}$ that scales the chiral contribution of the likelihood. We hypothesise that a good setting should account for the contiguity of motifs, but a principled choice is difficult to make a priori. For these reasons, TDS-fape and TDS-rmsd may be more suitable, as they work right out of the box.

In our attempt to scaffold two multi-motif problems with more samples, TDS-rmsd could only produce one unique success out of 512 generated backbones. Echoing our findings in the single-motif case, we attribute this to our fixed motif placements and the lack of sequence information in our inverse problem formulation, both of which are addressed by our baseline in Genie2. To put this in perspective, there are around one million possible motif placements for 1PRW_two, varying between 120 and 200 residues in total length. With the goal of demonstrating the feasibility of our setup, we used the same placements as our initial assessment, which, from our metrics, had hinted at its suitability. A variable motif placement is especially crucial for 3BIK+3BP5, as nearly all designs were grouped in the same cluster. Moreover, without accounting for sequence information, we may need to sample far more to cover the subset of backbones that can be attained by folding an amino acid chain containing all the motifs' sequences. This is evidenced by the higher number of successes among methods when the motif sequence is not fixed in the evaluation process (see App. D.2).

Despite our low success yield, we remain optimistic that future setups addressing our concerns can eventually narrow the gap between zero-shot and amortised scaffolding of multiple motifs, with our result as a demonstration of what is possible.

### 4.4 Symmetric Generation

Using replacement and reconstruction guidance methods, we tested our symmetric formulation by generating internally symmetric monomers for cyclic and dihedral point symmetries. Several met the designability criteria as shown in Fig. 5. Structures were less designable at higher orders of symmetry, as the subunits became increasingly short. Several designs were toroidal, with some resembling TIM barrels.

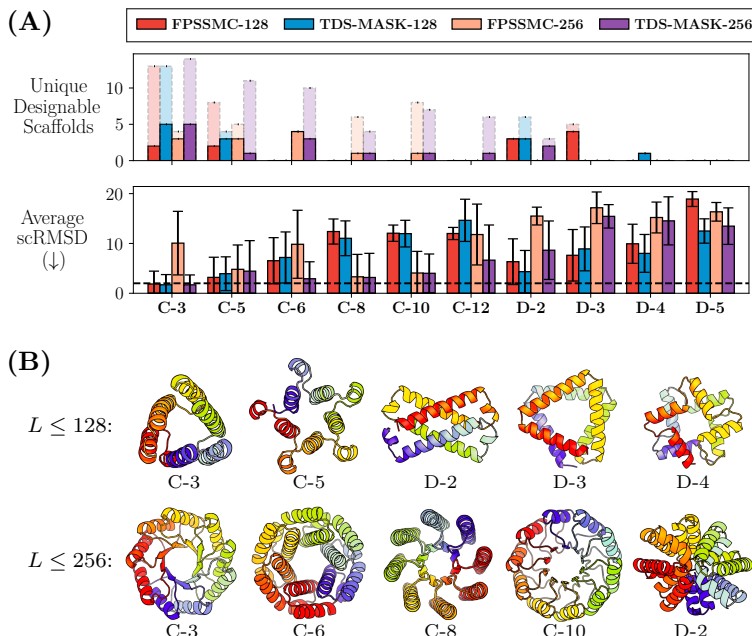

Figure 5: **(A) Designability of symmetric designs across several point symmetries**. Sixteen scaffolds with a maximum of 128 and 256 residues were sampled for each symmetry through FPSSMC and TDS-MASK. The total number of designable scaffolds is dashed atop the unique count. The success threshold for scRMSD is indicated by the dashed line. **(B) Examples of the successfully designed scaffolds.** The first and second rows show designs with a maximum of 128 and 256 residues, respectively. The primary axis of symmetry points directly outwards of the page.

While, our method implicitly imposes symmetry, the generated backbones were all found to be symmetric. We attribute this to the tight inverse problem variance and the samplers guiding the backbone in sufficiently meeting the inverse problem formulation. This implicit approach has some advantages over explicitly symmetrising backbones at each step. It allows for control over looser symmetries by increasing the variance $\sigma_v^2$. This widens the target space to include commonly observed monomeric proteins with non-exact internal symmetries. It is also composable with other constraints without having to orient the asymmetric subunits at fixed distances at each step.

### 4.5 Computational Costs

The samplers and guidance potentials used throughout our experiments incurred different computational costs. We outline their differences to inform future motif scaffolding experiments.

Table 1 summarises the cost and estimated runtime of each sampler. We report the number of forward and backward passes to the score model, as it was the main bottleneck of the methods. TDS was the most expensive, having to backpropagate through the entire score model each time, regardless of its guidance potential pairing. Empirically, this resulted in runtimes up to three times longer than those of the next most expensive sampler in MCGDiff, with the backward passes taking about twice as long as the forward pass. While the rest of the samplers rely solely on forward passes to the model, we found that a subtle design choice in their weight computations resulted in significant speed-ups between them. FPS-SMC and BPF with the noised measurements do not use the score model (i.e. to get $\mathbb{E}[\mathbf{x}_0 \mid \mathbf{x}_t]$ or $\mathbb{E}[\mathbf{x}_{t-1} \mid \mathbf{x}_t]$) when calculating the weights. This enables the simple optimisation of only evaluating the score of the unique particles after resampling. Noting that the ESS (estimator) can be interpreted as a rough estimate of the unique particle count, we found that this optimisation evaluates the score of only a fraction of the $K$ particles each time, roughly equal to the average ESS across all time steps $\widehat{\text{ESS}}_{\text{avg}}$. This led to speed-ups of up to $K$ times when the weights were consistently degenerate, a fairly common occurrence in a high-dimensional problem such as

motif scaffolding. For completeness, we also report the cost of BPF using the likelihood in (4). We did not evaluate it on the full benchmark, as its twisted version is a special case of TDS with a guidance scale of zero, and we found this to be sub-par to larger scale values in our hyperparameter search.

| Sampler | # Model Passes | | Est. Speedup over TDS |
|---|---|---|---|
| | FW | BW | |
| BPF (with noised $\{\mathbf{y}_t\}_{t=1}^T$) | $\approx \widehat{\mathrm{ESS}}_{\mathrm{avg}}T$ | 0 | $\frac{3K}{\widehat{\mathrm{ESS}}_{\mathrm{avg}}}$x |
| BPF (with approx. in (4)) | $KT$ | 0 | 3x |
| FPS-SMC | $\approx \widehat{\mathrm{ESS}}_{\mathrm{avg}}T$ | 0 | $\frac{3K}{\widehat{\mathrm{ESS}}_{\mathrm{avg}}}$x |
| MCGDiff | $KT$ | 0 | 3x |
| TDS | $KT$ | $KT$ | 1x |

Table 1: **The number of score model passes and estimated runtime speedup over TDS for each sampler.** Here, $K$ is the number of particles, $T$ is the number of denoising time steps, and $\widehat{\mathrm{ESS}}_{\mathrm{avg}}$ is the average ESS taken across all time steps.

| Potential | Runtime Complexity |
|---|---|
| Mask | $O(|\mathcal{M}|)$ |
| Distance | $O(|\mathcal{M}|^2)$ |
| Frame-Distance | $O(|\mathcal{M}|^2)$ |
| FAPE | $O(|\mathcal{M}|^2)$ |
| RMSD | $O(|\mathcal{M}|)$ |

Table 2: **The runtime complexity of guidance potential calculations in terms of motif length.**

The calculation of guidance potentials also contributes to the overall runtime, notably going through automatic differentiation, albeit considerably less than the score model invocations. How the runtime complexity scales with the motif length is listed in Table 2. Depending on the problem, the motif may have a total length more than half the entire protein (e.g. 1PRW). However, we found the quadratic complexity to be a non-issue given the relatively small maximum number of residues. In any case, masking, which maintains the problem's linearity, and RMSD, which has the most promising results, both scale linearly. With multiple motifs, the complexity is in terms of the longest motif.

# 5 Discussion

We proposed a set of guidance potentials and evaluated various SMC samplers for the motif scaffolding problem. We produced successful scaffolds for several single-motif and multi-motif problems in zero-shot. On single-motif problems, our proposed potentials perform comparably to masking while generalising to the multi-motif case, and reconstruction guidance outperformed replacement methods when aided by SMC. Internally symmetric monomers were also successfully designed by our setup, and their composition with motifs is left for subsequent work.

Our assessments were limited in the number of backbones sampled and motif placements considered. We welcome further exploration of the performance of various potentials with larger sample sizes and uniform sampling of valid motif placements. With scaffolding performance being only as good as the underlying unconditional model, the model-agnostic quality of the methods allows future, more performant models to be swapped in. Another major point of future work is incorporating sequence information in our setup to specify the correct target distribution and thereby improve the efficiency of samplers. Unlike structure, where bespoke potentials can be crafted, sequence information must likely come from pre-trained models with an inherent understanding of amino acid interactions. We have shown that it is possible to address multi-motif scaffolding without sequence information but believe it is essential to make zero-shot practices viable. Finally, several guidance potentials were motivated by work within the backbone modelling space and were rooted in the rigid body assumption involving idealised bond lengths and angles. With all-atom diffusion models on the horizon, additional work can be done to support these more general protein representations.

**Acknowledgements**

J.M.Y. would like to thank Joshua Southern for insightful discussions and a warm welcome to the world of protein design.

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

# A  Additional Background

## A.1  Diffusion Models and Protein Structure

**Denoising Diffusion Probabilistic Models (DDPMs).**  Recall that diffusion models learn to reverse a diffusion process applied to a target distribution $q(\mathbf{x}_0)$. Here, we outline them in more detail under the framework of denoising diffusion probabilistic models (DDPMs) (Ho et al., 2020). In general, their dynamics are governed by a *forward (diffusion)* process and a *reverse (denoising)* process—both of which are Markovian. Adopting the terminology of Ho et al. (2020), we start with the forward process with a transition kernel

$$q(\mathbf{x}_t \mid \mathbf{x}_{t-1}) := \mathcal{N}\left(\mathbf{x}_t;\ \sqrt{1-\beta_t}\mathbf{x}_{t-1},\ \beta_t\mathbf{I}\right), \tag{8}$$

for some decreasing variance schedule $\beta_1,\dots,\beta_T \in (0,1)$. Equivalently, denoting $\alpha_t := 1-\beta_t$ and $\bar{\alpha}_t := \prod_{s=1}^t \alpha_s$, the noising process can be done in a single step via

$$q(\mathbf{x}_t \mid \mathbf{x}_0) = \mathcal{N}\left(\mathbf{x}_t;\ \sqrt{\bar{\alpha}_t}\mathbf{x}_0,\ (1-\bar{\alpha}_t)\mathbf{I}\right). \tag{9}$$

For large enough $T$, we have $q(\mathbf{x}_T \mid \mathbf{x}_0) \approx \mathcal{N}(\mathbf{x}_T;\ \mathbf{0},\ \mathbf{I})$. As a result, the forward process constructs a bridge from the data distribution $q(\mathbf{x}_0)$ to the Gaussian distribution $\mathcal{N}(\mathbf{0},\mathbf{I})$. Naturally, the *time reversal* of this process maps Gaussian samples back to the data distribution. DDPMs do this by learning to measure the total noise accumulated from the forward process. Manipulating eq. (9) above, we can write

$$\mathbf{x}_0 = \frac{1}{\sqrt{\bar{\alpha}_t}}\left(\mathbf{x}_t - \sqrt{1-\bar{\alpha}_t}\epsilon_t\right), \quad \text{where } \epsilon_t \sim \mathcal{N}(\mathbf{0},\ \mathbf{I}), \tag{10}$$

which enables reconstructing the clean sample upon predicting the total noise. The reverse process, however, is done gradually rather than in a single step through the transition kernel

$$p_\theta(\mathbf{x}_{t-1} \mid \mathbf{x}_t) := \mathcal{N}\left(\mathbf{x}_{t-1};\ \mu_\theta(\mathbf{x}_t,t),\ \Sigma_\theta(t)\right),$$

parameterised by

$$\mu_\theta(\mathbf{x}_t,t) := \frac{\sqrt{\alpha_t}(1-\bar{\alpha}_{t-1})}{1-\bar{\alpha}_t}\mathbf{x}_t + \frac{\sqrt{\bar{\alpha}_{t-1}}\beta_t}{1-\bar{\alpha}_t}\hat{\mathbf{x}}_0(\mathbf{x}_t,t) \qquad\qquad \Sigma_\theta(t) := \beta_t\mathbf{I},$$
$$= \frac{1}{\sqrt{\alpha_t}}\left(\mathbf{x}_t - \frac{1-\alpha_t}{\sqrt{1-\bar{\alpha}_t}}\epsilon_\theta(\mathbf{x}_t,t)\right),$$

where $\epsilon_\theta(\mathbf{x}_t,t)$ is the total noise as predicted by a denoising network and $\hat{\mathbf{x}}_0(\mathbf{x}_t,t)$ is the predicted clean sample found by substituting $\epsilon_\theta(\mathbf{x}_t,t)$ into eq. (10). The procedure to generate samples $\mathbf{x}_0 \sim q(\cdot)$ then involves the reverse process

$$p_\theta(\mathbf{x}_{0:T}) = p(\mathbf{x}_T) \prod_{t=1}^T p_\theta(\mathbf{x}_{t-1} \mid \mathbf{x}_t),$$

first sampling from $\mathbf{x}_T \sim \mathcal{N}(\mathbf{0},\ \mathbf{I})$ then gradually denoising the samples for $T$ steps.

A key connection can be made with score-based generative models (SBGMs) (Song and Ermon, 2019) in their learning objectives. SBGMs predict the *score function* $s(\mathbf{x}) = \nabla_\mathbf{x}\log q(\mathbf{x})$, a vector that points in the direction of areas with high density in the data distribution. Taking the gradient of the log density of eq. (9), we precisely have

$$s_\theta(\mathbf{x}_t,t) = -\frac{\epsilon_\theta(\mathbf{x}_t,t)}{\sqrt{1-\bar{\alpha}_t}}.$$

When working with denoising networks, we use this relationship to compute the score.

**(A) Generic Protein Backbone**      **(B) Rigid Body Frame Representation**

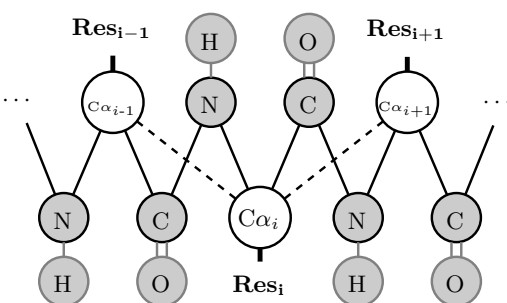
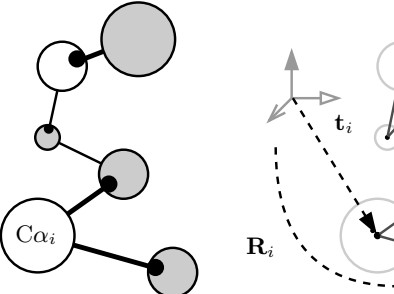

Figure 6: **The protein backbone illustrated**. **(A)** The backbone is linear, with a repeating $N - C\alpha - C$ atomic structure. Each side chain residue is anchored at a $C\alpha$ atom, varying structurally depending on its amino acid type. **(B)** Fixing the $N - C\alpha - C$ substructures as rigid bodies, the $i$th residue can be represented as a (triangular) frame, defined by a rotation matrix $\mathbf{R}_i$ and a translation vector $\mathbf{t}_i$ with respect to a global reference frame.

**Primer on Protein Structure.** A protein's *tertiary* structure—its three-dimensional arrangement in space—is often the modality of choice for generative modelling. A key reason is that applications are centred around designing proteins with some function, and while the sequence defines the protein, its function is more evident in its structure. Fig. 6 illustrates this structure, which comprises the backbone and side chains. Like most generative efforts, we limit our scope to monomeric (i.e., single-chain) protein backbones. Only after the backbone is modelled are the side chains predicted.

Since protein backbones are high-dimensional, an efficient representation incorporating their symmetries is ideal for making their associated learning problems more tractable. First, we may assume idealised bond lengths and angles, allowing the $C\alpha$ coordinates to be representative of the entire backbone. Next, we note that rotations and translations in three-dimensional space do not change the inherent structure of the protein. These symmetries, as well as the *chiral* properties of proteins, associate them with the *three-dimensional special Euclidean group*, SE(3), where each transformation $\mathbf{T} = (\mathbf{R}, \mathbf{t}) \in \mathrm{SE}(3)$ is a pair of a rotation matrix $\mathbf{R} \in \mathbb{R}^{3 \times 3}$ and a translation vector $\mathbf{t} \in \mathbb{R}^3$. Given the $C\alpha$ coordinates $\mathbf{x} \in \mathbb{R}^{N \times 3}$ of a protein backbone with $N$ residues, we denote $\mathbf{T} \circ \mathbf{x}$ as the backbone resulting from the transformation, where $(\mathbf{T} \circ \mathbf{x})_i = \mathbf{R}\mathbf{x}_i + \mathbf{t}$.

The *frame* representation (Jumper et al., 2021) incorporates the above symmetry group by treating each $N - C\alpha - C$ substructure as a rigid body, whose orientation and position are indicated by a transformation in SE(3) with respect to a global reference frame. Here, we denote the $i$th residue of the backbone as $\mathbf{T}_i(\mathbf{x}) = (\mathbf{R}_i(\mathbf{x}), \mathbf{t}_i(\mathbf{x})) \in \mathrm{SE}(3)$, derived from the $C\alpha$ coordinates $\mathbf{x}$ via the Gram-Schmidt process (Alg. 2). This formulation is used by several diffusion models (Watson et al., 2023; Yim et al., 2023; Lin et al., 2024) to learn the distribution of protein backbones efficiently.

---

**Algorithm 2:** Frame Construction from Atom Coordinates (Adapted from Supplementary Material Algorithm 21 of Jumper et al. (2021))

---

**input** : coordinates of $i$th N, C-$\alpha$, and C atoms $\mathbf{x}_{i,N}, \mathbf{x}_{i,CA}, \mathbf{x}_{i,C}$
**output**: frame representation of $i$th residue $(\mathbf{R}_i, \mathbf{t}_i)$
`# Get vectors pointing from C-α to N and C`
Set $\mathbf{v}_{i,1} \leftarrow \mathbf{x}_{i,C} - \mathbf{x}_{i,CA}$
Set $\mathbf{v}_{i,2} \leftarrow \mathbf{x}_{i,N} - \mathbf{x}_{i,CA}$
`# Do Gram-Schmidt process`
Set $\mathbf{e}_{i,1} \leftarrow \mathbf{v}_{i,1}/\|\mathbf{v}_{i,1}\|$
Set $\mathbf{u}_{i,2} \leftarrow \mathbf{v}_{i,2} - \mathbf{e}_{i,1}\left(\mathbf{e}_{i,1}^\top \mathbf{v}_{i,2}\right)$
Set $\mathbf{e}_{i,2} \leftarrow \mathbf{u}_{i,2}/\|\mathbf{u}_{i,2}\|$
Set $\mathbf{e}_{i,3} \leftarrow \mathbf{e}_{i,1} \times \mathbf{e}_{i,2}$
`# Construct frame components`
Set $\mathbf{R}_i \leftarrow (\mathbf{e}_{i,1} \mid \mathbf{e}_{i,2} \mid \mathbf{e}_{i,3})$
Set $\mathbf{t}_i \leftarrow \mathbf{x}_{i,CA}$

---

## A.2 SMC-Aided Diffusion Posterior Samplers

With SMC, we reiterate that two main design choices are available: the proposals $q_t$ and the intermediate targets $\gamma_t$. In this section, we detail the specific choices made by the SMC samplers considered in our experiments. We first outline how a linear structure to the inverse problem can define a noisy measurement process. This is utilised by two existing samplers, FPS-SMC and MCGDiff, that operate on linear problems. Then, we discuss how each sampler fits the recipe in Alg. 1. Seeing as the weight updates can be a source of confusion, we show their derivations. Where there are distributions that we are free to design, we time-index them.

**Linear Measurement Process.** We follow the construction by Dou and Song (2023) for producing a measurement process under a linear inverse problem

$$\mathbf{y}_0 = \mathbf{A}\mathbf{x}_0 + \sigma_v \mathbf{z}_0,$$

where $\mathbf{y}_0 \in \mathbb{R}^d$, $\mathbf{x}_0 \in \mathbb{R}^D$, $\mathbf{A} \in \mathbb{R}^{d \times D}$, and $\mathbf{z}_0 \sim \mathcal{N}(\mathbf{0}, \mathbf{I})$. Recall that the DDPM's forward process (8) can be written as

$$\mathbf{x}_t = \sqrt{1 - \beta_t}\mathbf{x}_{t-1} + \sqrt{\beta_t}\mathbf{z}_{t-1},$$

for $\mathbf{z}_{t-1} \sim \mathcal{N}(\mathbf{0}, \mathbf{I})$. Applying the matrix $\mathbf{A}$ to this recursion, we can define an analogous measurement process

$$\mathbf{y}_t = \sqrt{1 - \beta_t}\mathbf{y}_{t-1} + \sqrt{\beta_t}\mathbf{A}\mathbf{z}_{t-1},$$

where $\mathbf{z}_{t-1}$ is randomly sampled or chosen to be equal to the noise applied in the latent forward process. The intermediate noisy measurements can then be constructed by

$$\nu(\mathbf{y}_{1:T} \mid \mathbf{y}_0) = \prod_{t=1}^{T} \nu(\mathbf{y}_t \mid \mathbf{y}_{t-1}), \tag{11}$$
$$\text{where } \nu(\mathbf{y}_t \mid \mathbf{y}_{t-1}) = \mathcal{N}(\mathbf{y}_t; \ \sqrt{1 - \beta_t}\mathbf{y}_{t-1}, \ \beta_t \mathbf{A}\mathbf{A}^\top).$$

Using the above recursions and the inverse problem statement, the likelihood has the form

$$g(\mathbf{y}_t \mid \mathbf{x}_t) = \mathcal{N}(\mathbf{y}_t; \ \mathbf{A}\mathbf{x}_t, \ \sigma_v^2 \bar{\alpha}_t \mathbf{I}), \tag{12}$$

and its logarithm's gradient is precisely the guidance term for replacement methods. From here, we can develop a simple particle filter by considering BPF, the baseline sampler for SSMs.

**Bootstrap Particle Filter.** For now, let us assume a (filtering) model with sequential measurements and the likelihood in eq. (12). In BPF, the proposal $q_t$ matches the DDPM's reverse kernel $p_\theta$ and the target $\gamma_t$ is the joint distribution between the latent and measurement variables. Under this choice, the weight update becomes

$$\tilde{w}_t = \frac{p_\theta(\mathbf{x}_t \mid \mathbf{x}_{t+1}) g_t(\mathbf{y}_t \mid \mathbf{x}_t)}{p_\theta(\mathbf{x}_t \mid \mathbf{x}_{t+1})} = \boxed{g_t(\mathbf{y}_t \mid \mathbf{x}_t)},$$

and the likelihood values essentially become the fitness criteria for filtering samples.

To improve the efficiency of BPF, we can choose $q_t$ to be the locally *optimal proposal* $q_t^*$, better estimating the target with fewer particles. This notion is formalised by minimising the KL-divergence between $\gamma_{t+1}(\mathbf{x}_{t+1:T}) q_t(\mathbf{x}_t \mid \mathbf{x}_{t+1})$ and $\gamma_t(\mathbf{x}_{t:T})$. For SSMs, this is a known result with

$$q_t^*(\mathbf{x}_t \mid \mathbf{x}_{t+1}, \mathbf{y}_t) \propto \underbrace{\frac{\gamma_t(\mathbf{x}_{t:T})}{\gamma_{t+1}(\mathbf{x}_{t+1:T})}}_{\tilde{f}_t(\mathbf{x}_t \mid \mathbf{x}_{t+1})} = p_\theta(\mathbf{x}_t \mid \mathbf{x}_{t+1}) g_t(\mathbf{y}_t \mid \mathbf{x}_t),$$

where $\tilde{f}_t$ is what we refer to as the transition between targets. Note that this can be different for a terminal measurement model, where we do not need to explain each intermediate measurement, only the last one.

**Filtering Posterior Sampling.** Dou and Song (2023) precisely leverage the linear measurement process and the corresponding optimal proposal. As the likelihood in eq. (12) and the reverse diffusion kernel are both Gaussian, the optimal proposal $q_t^*$ is analytically available and is given by

$$q_t^*(\mathbf{x}_t \mid \mathbf{x}_{t+1}, \mathbf{y}_t) = \mathcal{N}(\mathbf{x}_t;\ \mu^*(\mathbf{x}_{t+1}, \mathbf{y}_t, t+1),\ \Sigma^*(t+1)), \tag{13}$$

$$\Sigma^*(t) = \left(\Sigma_\theta(t)^{-1} + \frac{1}{\sigma_v^2 \bar{\alpha}_{t-1}} \mathbf{A}^\top \mathbf{A}\right)^{-1},$$

$$\mu^*(\mathbf{x}_t, \mathbf{y}_{t-1}, t) = \Sigma^*(t)\left(\Sigma_\theta(t)^{-1} \mu_\theta(\mathbf{x}_t, t) + \frac{1}{\sigma_v^2 \bar{\alpha}_{t-1}} \mathbf{A}^\top \mathbf{y}_{t-1}\right).$$

With the optimal proposal, the new weight update is simply

$$\boxed{\tilde{w}_t = \frac{p_\theta(\mathbf{x}_t \mid \mathbf{x}_{t+1}) g_t(\mathbf{y}_t \mid \mathbf{x}_t)}{q_t^*(\mathbf{x}_t \mid \mathbf{x}_{t+1}, \mathbf{y}_t)}}.$$

However, instead of forward-noising $\mathbf{y}_0$, they find improved performance through a noise-sharing technique by setting $\mathbf{y}_T = \mathbf{A}\mathbf{x}_T$ and building the sequence backwards with

$$\nu_{FPS}(\mathbf{y}_{t-1} \mid \mathbf{y}_t, \mathbf{y}_0) = \mathcal{N}\left(\mathbf{y}_{t-1};\ \sqrt{\bar{\alpha}_{t-1}}\mathbf{y}_0 + \sqrt{\frac{(1-c)(1-\bar{\alpha}_{t-1})}{1-\bar{\alpha}_t}}(\mathbf{y}_t - \sqrt{\bar{\alpha}_t}\mathbf{y}_0),\ c(1-\bar{\alpha}_{t-1})\mathbf{A}^\top \mathbf{A}\right)$$

for some tunable parameter $c \in [0, 1]$. We choose $c = \beta_t/(1 - \bar{\alpha}_{t-1})$ to match our DDPM variance $\Sigma_\theta(t) = \beta_t \mathbf{I}$.

**Monte Carlo Guided Diffusion.** Cardoso et al. (2023) use the same optimal proposal $q_t^*$ in eq. (13) (for sequential measurements) but adopt a terminal measurement model. To improve efficiency, they tilt the targets with the twist function $\psi_t = g_t(\mathbf{y}_t \mid \mathbf{x}_t)$. Because of their algorithm's construction, where the final samples are not reweighted (i.e. $\tilde{w}_0 = 1$), the weight updates are shifted one time step in advance. For $t > 1$,

$$\tilde{w}_t = \frac{p_\theta(\mathbf{x}_{t-1} \mid \mathbf{x}_t)}{q_t^*(\mathbf{x}_{t-1} \mid \mathbf{x}_t, \mathbf{y}_{t-1})} \frac{g_{t-1}(\mathbf{y}_{t-1} \mid \mathbf{x}_{t-1})}{g_t(\mathbf{y}_t \mid \mathbf{x}_t)} = \frac{p_\theta(\mathbf{x}_{t-1} \mid \mathbf{x}_t)}{\frac{p_\theta(\mathbf{x}_{t-1} \mid \mathbf{x}_t) g_{t-1}(\mathbf{y}_{t-1} \mid \mathbf{x}_{t-1})}{\int p_\theta(\mathbf{x}_{t-1} \mid \mathbf{x}_t) g_{t-1}(\mathbf{y}_{t-1} \mid \mathbf{x}_{t-1}) d\mathbf{x}_{t-1}}} \frac{g_{t-1}(\mathbf{y}_{t-1} \mid \mathbf{x}_{t-1})}{g_t(\mathbf{y}_t \mid \mathbf{x}_t)},$$

$$= \frac{\int p_\theta(\mathbf{x}_{t-1} \mid \mathbf{x}_t) g_{t-1}(\mathbf{y}_{t-1} \mid \mathbf{x}_{t-1}) d\mathbf{x}_{t-1}}{g_t(\mathbf{y}_t \mid \mathbf{x}_t)} \approx \boxed{\frac{g_{t-1}(\mathbf{y}_{t-1} \mid \hat{\mathbf{x}}_{t-1})}{g_t(\mathbf{y}_t \mid \mathbf{x}_t)}}, \tag{14}$$

where we used $\hat{\mathbf{x}}_{t-1} = \mathbb{E}[\mathbf{x}_{t-1} \mid \mathbf{x}_t]$ as a single-sample approximation of the integral. Note that, at $t = 1$, we get the same weight computation, as $\psi_0 = 1$ but the target transition contains an extra term $g_0(\mathbf{y}_0 \mid \mathbf{x}_0)$ to explain the terminal measurement. To maintain the correct targets in the midst of the shifted weight computations, the first weight is set to be $\tilde{w}_T = g_{T-1}(\mathbf{y}_{T-1} \mid \hat{\mathbf{x}}_{T-1})$, where $\hat{\mathbf{x}}_{T-1} = \mathbb{E}[\mathbf{x}_{T-1} \mid \mathbf{x}_T]$.

Furthermore, instead of noising $\mathbf{y}_0$, they set $\mathbf{y}_t = \sqrt{\bar{\alpha}_t}\mathbf{y}_0$, the conditional expectation of the measurement process given the terminal measurement. For experiments, we use the noiseless inpainting version of MCGDiff (i.e. $\sigma_v = 0$ and $\mathbf{A}$ is a masking matrix), which involves a similar proposal, but we omit the details here.

**Twisted Diffusion Sampler.** Wu et al. (2023) similarly adopt a terminal measurement model, with the twist function $\psi_t = g_t(\mathbf{y}_0 \mid \hat{\mathbf{x}}_0(\mathbf{x}_t, t))$ and proposal $q_t^{TDS}$ relying on the likelihood approximation in eq. (4). Their proposal is given by

$$q_t^{TDS}(\mathbf{x}_t \mid \mathbf{x}_{t+1}, \mathbf{y}_0) = \mathcal{N}\left(\mathbf{x}_t;\ \frac{1}{\sqrt{\bar{\alpha}_t}}(\mathbf{x}_{t+1} + (1 - \alpha_t)s_{t+1}(\mathbf{x}_{t+1}, \mathbf{y}_0, t+1)),\ \Sigma_\theta(t)\right), \tag{15}$$

where we have the conditional score approximation

$$s_t(\mathbf{x}_t, \mathbf{y}_0, t) = s_\theta(\mathbf{x}_t, t) + \nabla_{\mathbf{x}_t} \log g(\mathbf{y}_0 \mid \hat{\mathbf{x}}_0(\mathbf{x}_t, t)).$$

For $t > 0$, this leads to the weight update

$$\tilde{w}_t = \boxed{\frac{p_\theta(\mathbf{x}_t \mid \mathbf{x}_{t+1})}{q_t^{TDS}(\mathbf{x}_t \mid \mathbf{x}_{t+1}, \mathbf{y}_0)} \frac{g_t(\mathbf{y}_0 \mid \hat{\mathbf{x}}_0(\mathbf{x}_t, t))}{g_{t+1}(\mathbf{y}_0 \mid \hat{\mathbf{x}}_0(\mathbf{x}_{t+1}, t+1))}},$$

which also coincides with the update for $t = 0$, as $\psi_0 = 1$ and the terminal measurement has to be explained.

| Sampler | Proposal ($q_t$) | Twist Function ($\psi_t$) | Measurement Model |
|---------|------------------|---------------------------|-------------------|
| BPF | $p_\theta(\mathbf{x}_t \mid \mathbf{x}_{t+1})$ | $1$ | Any |
| FPS-SMC | $q_t^*(\mathbf{x}_t \mid \mathbf{x}_{t+1}, \mathbf{y}_t)$ (eq. (13)) | $1$ | Sequential |
| MCGDiff | $q_t^*(\mathbf{x}_t \mid \mathbf{x}_{t+1}, \mathbf{y}_t)$ (eq. (13)) | $g_t(\mathbf{y}_t \mid \mathbf{x}_t)$ | Terminal |
| TDS | $q_t^{TDS}(\mathbf{x}_t \mid \mathbf{x}_{t+1}, \mathbf{y}_0)$ (eq. (15)) | $g_t(\mathbf{y}_0 \mid \hat{\mathbf{x}}_0(\mathbf{x}_t, t))$ | Terminal |

Table 3: **The design choices of each SMC sampler for diffusion posterior sampling.** To define their intermediate measurements, FPS-SMC samples from the measurement noising process (11), and MCGDiff takes the expectation of the same process conditioned on the terminal measurement.

We have shown that these existing works are merely differentiated by their design choices. Table 3 gives an overview. While they all target the same distribution under the right assumptions, their efficiencies, or lack thereof, are more prominent in the finite particle setting. Even more, the high cost of the score model can impose significantly smaller particle counts. The high-dimensional nature of inverse problems, such as motif scaffolding, compounds the effect so that the weights become almost always degenerate. As a consequence, virtually all samplers will return multiple copies of the same sample. This is because particles have less room to diversify towards the end of the reverse process, and resampling in the later stages becomes detrimental to diversity. Some workarounds include resampling only until a certain point and running parallel instances of samplers. In any case, utilising the optimal proposal and twisted targets has become commonplace in the most performant samplers.

# B    Likelihood Formulation

### B.1    $\mathrm{SE}(3)$-**Invariance and Chirality of Frame-Distance Likelihood**

The potential in eq. (7) has distance and chiral components—both of which are translation invariant. It remains to be shown that the chiral component is invariant to rotations but not reflections.

Suppose we have a protein's three-dimensional C-$\alpha$ coordinates $\mathbf{x}$. It can be converted into a set of frames $\{(\mathbf{R}_i, \mathbf{t}_i)\}_{i=1}^L$ via Alg. 2. Note that the positions for the N and C atoms are fixed, given the rigid body assumption, and are determined based on the adjacent C-$\alpha$ coordinates. We consider the case of a reflected protein $\mathbf{x}_{\mathrm{ref}} := -\mathbf{x}$ and a rotated protein $\mathbf{x}_{\mathrm{rot}} := \mathbf{R}_\theta \mathbf{x}$. Under the Gram-Schmidt process, we get the frame representations

$$\mathbf{x}_{\mathrm{ref},i} \mapsto \left(\mathbf{R}_i \left(\begin{smallmatrix} -1 & 0 & 0 \\ 0 & -1 & 0 \\ 0 & 0 & 1 \end{smallmatrix}\right), \ -\mathbf{t}_i\right), \quad \mathbf{x}_{\mathrm{rot},i} \mapsto (\mathbf{R}_\theta \mathbf{R}_i, \ \mathbf{R}_\theta \mathbf{t}_i).$$

First, the pairwise deviations between each residue are computed. For a pair $(i, j)$, we have

$$\mathbf{R}(\mathbf{x}_{\mathrm{ref},i})^\top \mathbf{R}(\mathbf{x}_{\mathrm{ref},j}) = \left(\mathbf{R}_i \left(\begin{smallmatrix} -1 & 0 & 0 \\ 0 & -1 & 0 \\ 0 & 0 & 1 \end{smallmatrix}\right)\right)^\top \mathbf{R}_j \left(\begin{smallmatrix} -1 & 0 & 0 \\ 0 & -1 & 0 \\ 0 & 0 & 1 \end{smallmatrix}\right) = \left(\begin{smallmatrix} -1 & 0 & 0 \\ 0 & -1 & 0 \\ 0 & 0 & 1 \end{smallmatrix}\right) \mathbf{R}_i^\top \mathbf{R}_j \left(\begin{smallmatrix} -1 & 0 & 0 \\ 0 & -1 & 0 \\ 0 & 0 & 1 \end{smallmatrix}\right),$$

$$\mathbf{R}(\mathbf{x}_{\mathrm{rot},i})^\top \mathbf{R}(\mathbf{x}_{\mathrm{rot},j}) = (\mathbf{R}_\theta \mathbf{R}_i)^\top \mathbf{R}_\theta \mathbf{R}_j = \mathbf{R}_i^\top \mathbf{R}_\theta^\top \mathbf{R}_\theta \mathbf{R}_j = \mathbf{R}_i^\top \mathbf{R}_j,$$

which shows that such a deviation, and therefore the chiral component as a whole, is invariant to rotations. We remark that this expression is different from $\mathbf{R}_i \mathbf{R}_j^\top$, the rotation matrix that transforms the $j$th frame's orientation to that of the $i$th frame. Next, the cosine of the deviations' angles are computed. In the reflection case, we have

$$d_{\cos}\big(\mathbf{R}_i^\top \mathbf{R}_j, \ \mathbf{R}(\mathbf{x}_{\mathrm{ref},i})^\top \mathbf{R}(\mathbf{x}_{\mathrm{ref},j})\big) = \frac{1}{2}\left(\mathrm{Tr}\left(\mathbf{R}_i^\top \mathbf{R}_j \left(\begin{smallmatrix} -1 & 0 & 0 \\ 0 & -1 & 0 \\ 0 & 0 & 1 \end{smallmatrix}\right) \mathbf{R}_j^\top \mathbf{R}_i \left(\begin{smallmatrix} -1 & 0 & 0 \\ 0 & -1 & 0 \\ 0 & 0 & 1 \end{smallmatrix}\right)\right) - 1\right).$$

Setting $\mathbf{V} = \mathbf{R}_i^\top \mathbf{R}_j$ and expanding the expression, we have

$$= \frac{1}{2}\left(\mathrm{Tr}(\mathbf{V}\mathbf{V}^\top) - 2(v_{13}^2 + v_{23}^2 + v_{31}^2 + v_{32}^2) - 1\right)$$

$$= 1 - (v_{13}^2 + v_{23}^2 + v_{31}^2 + v_{32}^2),$$

where we have used the fact that $\mathrm{Tr}(\mathbf{V}\mathbf{V}^\top) = \mathrm{Tr}(\mathbf{I}) = 3$ for rotation matrix $\mathbf{V}$. For the above to be equal to one, i.e. the cosine of zero, we must have $v_{13} = v_{23} = v_{31} = v_{32} = 0$ which precisely requires $\mathbf{R}_i^\top \mathbf{R}_j$ to be a rotation matrix strictly about the $z$-axis—which is not true in general. Hence, the orientation component is not generally reflection-invariant.

## C   Experimental Setup

### C.1   Benchmark Problems

In the following benchmarks, we fix the motif placement by taking the median of scaffold length ranges in their specifications like Wu et al. (2023). We isolate this variability to provide a less stochastic comparison of the methods over small sample sizes.

**Motif Scaffolding Benchmark.**   Problems in the RFDiffusion motif scaffolding benchmark are listed in Table 4. Excluding 6VW1, which involves multiple chains, there are 24 problems involving different motif types, lengths, and contiguity. Where the length and configuration define a range, we can design any length scaffold that fits those specifications but choose to fix the configuration in our experiments.

| Name | Description | Configuration | Length |
|------|-------------|---------------|--------|
| 1PRW | Double EF-hand motif | 5-20, **A16-35**, 10-25, **A52-71**, 5-20 | 60-105 |
| 1BCF | Di-iron binding motif | 8-15, **A92-99**, 16-30, **A123-130**, 16-30, **A47-54**, 16-30, **A18-25**, 8-15 | 96-152 |
| 5TPN | RSV F-protein Site V | 10-40, **A163-181**, 10-40 | 50-75 |
| 5IUS | PD-L1 binding interface on PD-1 | 0-30, **A119-140**, 15-40, **A63-82**, 0-30 | 57-142 |
| 3IXT | RSV F-protein Site II | 10-40, **P254-277**, 10-40 | 50-75 |
| 5YUI | Carbonic anhydrase active site | 5-30, A**93-97**, 5-20, **A118-120**, 10-35, **A198-200**, 10-30 | 50-100 |
| 1QJG | Delta5-3-ketosteroid isomerase active site | 10-20, **A38**, 15-30, **A14**, 15-30, **A99**, 10-20 | 53-103 |
| 1YCR | P53 helix that binds to Mdm2 | 10-40, **B19-27**, 10-40 | 40-100 |
| 2KL8 | De novo designed protein | **A1-7**, 20, **A28-79** | 79 |
| 7MRX_60 | Barnase ribonuclease inhibitor | 0-38, **B25-46**, 0-38 | 60 |
| 7MRX_85 | Barnase ribonuclease inhibitor | 0-68, **B25-46**, 0-63 | 85 |
| 7MRX_128 | Barnase ribonuclease inhibitor | 0-122, **B25-46**, 0-122 | 128 |
| 4JHW | RSV F-protein Site 0 | 10-25, **F196-212**, 15-30, **F63-69**, 10-25 | 60-90 |
| 4ZYP | RSV F-protein Site 4 | 10-40, **A422-436**, 10-40 | 30-50 |
| 5WN9 | RSV G-protein 2D10 site | 10-40, **A170-189**, 10-40 | 35-50 |
| 5TRV_short | De novo designed protein | 0-35, **A45-65**, 0-35 | 56 |
| 5TRV_med | De novo designed protein | 0-65, **A45-65**, 0-65 | 86 |
| 5TRV_long | De novo designed protein | 0-95, **A45-65**, 0-95 | 116 |
| 6E6R_short | Ferridoxin Protein | 0-35, **A23-35**, 0-35 | 48 |
| 6E6R_med | Ferridoxin Protein | 0-65, **A23-35**, 0-65 | 78 |
| 6E6R_long | Ferridoxin Protein | 0-95, **A23-35**, 0-95 | 108 |
| 6EXZ_short | RNA export factor | 0-35, **A28-42**, 0-35 | 50 |
| 6EXZ_med | RNA export factor | 0-65, **A28-42**, 0-65 | 80 |
| 6EXZ_long | RNA export factor | 0-95, **A28-42**, 0-95 | 110 |

Table 4: **RFDiffusion motif scaffolding benchmark details.** The specification for each scaffolding problem is under "Configuration", with the motif structures in bold. For example, in motif 2KL8, the problem requires a protein that contains residues 1-7 and 28-79 from chain A of the motif, joined together by a scaffold of 20 residues. Furthermore, the total length of the generated protein must fall in the range specified in the "Length" column.

**Multi-Motif Scaffolding Benchmark.**   Problems in the Genie2 multi-motif scaffolding benchmark are listed in Table 5. In the Genie2 pre-print, problem 3NTN had a configuration with ranges in reverse, but the specification differed from their GitHub repository. We chose to work with the ranges specified in their repository and made the correction in Table 5.

| Name | Description | Configuration | Length |
|------|-------------|---------------|--------|
| 4JHW+5WN9 | Two epitopes | 10-40, **4JHW/F254-278{1}**, 20-50, **5WN9/A170-189{2}**, 10-40 | 85-175 |
| 1PRW_two | Two 4-helix bundles | 5-20, **1PRW/A16-35{1}**, 10-25, **1PRW/A52-71{1}**, 10-30, **1PRW/A89-108{2}**, 10-25, **1PRW/A125-144{2}**, 5-20 | 120-200 |
| 1PRW_four | Four EF-hands | 5-20, **1PRW/A21-32{1}**, 10-25, **1PRW/A57-68{2}**, 10-25, **1PRW/A94-105{3}**, 10-25, **1PRW/A125-144{4}**, 5-20 | 88-163 |
| 3BIK+3BP5 | Two PD-1 binding motifs | 5-15, **3BIK/A121-125{1}**, 10-20, **3BP5/B110-114{2}**, 5-15 | 30-60 |
| 3NTN | Two 3-helix bundles | **3NTN/A342-348{1}**, 10, **3NTN/A367-372{2}**, 10-20, **3NTN/B342-348{2}**, 10, **3NTN/B367-372{1}**, 10-20, **3NTN/C367-372{1}**, 10, **3NTN/C342-348{2}** | 89-109 |
| 2B5I | Two binding sites | 5-15, **2B5I/A11-23{2}**, 10-20, **2B5I/A35-45{1}**, 10-20, **2B5I/A61-72{1}**, 5-15, **2B5I/A81-95{2}**, 20-30, **2B5I/A119-133{2}** | 116-166 |

Table 5: **Genie2 multi-motif scaffolding benchmark details.** The specification for each scaffolding problem is under "Configuration", with the motif structures in bold. Here, the motif structures are formatted as `<MOTIF_NAME>/<CHAIN_SEGMENT>{<MOTIF_GROUP>}`, where structures belonging to the same motif group are fixed in their orientations relative to each other. Furthermore, the total length of the generated protein must fall in the range specified in the "Length" column.

## C.2 Diffusion Posterior Samplers

By defining the likelihoods appropriately, we can use the guidance potentials together with the SMC samplers. For FPS-SMC and MCGDiff, which operate on linear inverse problems only, they are confined to using the masking potential, and thereby the likelihood

$$g_t(\mathbf{y}_t \mid \mathbf{x}_t) \propto \exp\left(-\frac{1}{\eta_t} L_{\text{mask}}\left(\mathbf{x}_t; \mathbf{y}_t, \mathcal{M}\right)\right),$$

with $\eta_t = 1/2\sigma_v^2 \bar{\alpha}_t$ for FPS-SMC and, for MCGDiff, $\eta_t = 1/(1-\bar{\alpha}_t)$ when $\mathbf{x}_t$ is known and $\eta_t = 1/(\beta_{t+1}^2 + 1 - \bar{\alpha}_t)$ when using $\hat{\mathbf{x}}_t = \mathbb{E}[\mathbf{x}_t \mid \mathbf{x}_{t+1}]$ in the numerator of eq. (14). On the other hand, TDS uses the DPS likelihood approximation and adopts a general likelihood

$$g_t(\mathbf{y}_0 \mid \mathbf{x}_t) \propto \exp\left(-\frac{1}{\eta_t} L\left(\hat{\mathbf{x}}_0(\mathbf{x}_t, t); \mathbf{y}_t, \mathcal{M}\right)\right),$$

where $\eta_t = 1/\tilde{\sigma}_t^2$. Wu et al. (2023) recommend setting $\tilde{\sigma}_t^2 := \text{Var}[\mathbf{x}_t \mid \mathbf{x}_0]$, leniently filtering early in the reverse process, when the reconstructed sample $\hat{\mathbf{x}}_0$ is still unreliable, and gradually tightening the filter towards the end. To keep it non-zero we set $\tilde{\sigma}_t^2 = (1 - \bar{\alpha}_t) + \sigma_v^2$ to match our inverse problem formulation at $t = 0$.

In our implementation of the samplers, we make several optimisations. By partitioning particles into groups, we can run multiple trials simultaneously, making the most of throughput gains with increased model batch sizes. This is achieved by computing weights and resampling on a per-group basis. We further perform multiprocessing across several GPUs. Beyond parallelism, we minimise the number of calls made to the diffusion model—the main bottleneck. We cache the predicted noise (or score) between computing weights and sampling from the proposal. Moreover, we only compute the predicted noise of unique particles for BPF and FPS-SMC, where duplicate particles may be fed to the proposal. Different batches of Gaussian noise are added afterwards to differentiate the duplicate particles. Computationally, it is as if we set the number of particles equal to the mean of the ESS (estimator).

# D   Additional Results

## D.1   Hyperparameter Search

The hyperparameters of each method were chosen to maximise performance on two selected motif problems, 1PRW and 3IXT. In a discretised grid-search fashion, 16 backbones are sampled and evaluated for each parameter combination. A key hyperparameter we consider is the diffusion model's *noise* or *temperature scale* $\zeta \in (0, 1]$ that scales the noise added to each step in the reverse process. A high $\zeta$ leads to higher sample diversity, and a low $\zeta$ typically yields higher sample quality. The Genie models have been shown to attain the best F1 designability-diversity score for unconditional generation at around $\zeta = 0.4$ (Lin and Alquraishi, 2023). However, the effect of this setting on the conditional samplers wrapped around the model is unclear. We tested three values $\zeta = 0.4, 0.7, 1.0$ to gauge the impact on sampler performance. The proposals of the samplers, all of which were Gaussian, had their standard deviation scaled by this value.

Fig. 7 shows that, for the replacement methods, lower values indeed improved the scRMSD but had different effects on motif RMSD depending on the motif. We suspect that with minimal noise, motifs common to backbones lying in the modes of the data distribution are easily scaffolded, whereas those outside are not. We chose $\zeta = 0.4$ for both replacement methods to prioritise a lower scRMSD, given the motif RMSD was already near the success threshold.

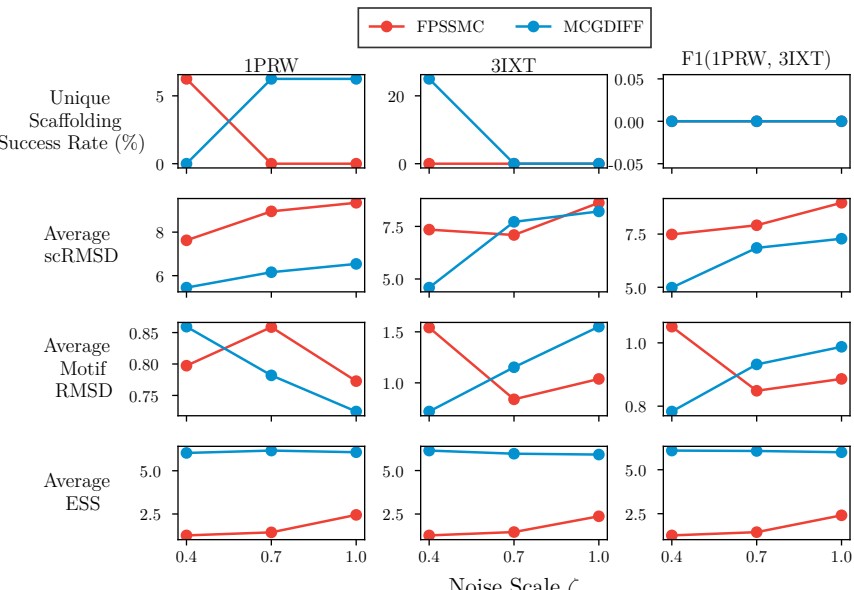

Figure 7: **Noise scale grid search for FPSSMC and MCGDIFF**. The unique success rate and average values for two of the four designability criteria are shown. The right column depicts the F1 score between the values in the left and middle columns.

In reconstruction guidance, an additional hyperparameter we consider is the guidance scale $\gamma$ that controls the strength of the conditional signal. For high $\gamma$, the target distribution becomes more peaky, and the adherence to the conditional label is higher at the cost of diversity. We consider six values in $\gamma = 0, 0.25, 0.5, 1.0, 2.0, 4.0$. Notably, $\gamma = 0$ is identical to BPF under a likelihood involving the reconstructed sample and the label. Fig. 8 shows results for TDS-MASK and TDS-DIST. For TDS-MASK, we chose $\gamma = 1.0$ and $\zeta = 1.0$, one of the two with the best unique success rate but with the lower average scRMSD. Owing to the reflection-invariance of TDS-DIST, higher guidance scales made it produce reflected motifs up to 50% of the time. We found this issue less pervasive with a smaller guidance scale, where the unconditional model, informed on handedness, makes a bigger contribution to the de-noising process. We therefore configured TDS-DIST at $\gamma = 0.25$ and $\zeta = 0.4$. As for TDS-FAPE and TDS-RMSD, there were no issues whatsoever with reflected motifs. Fig. 9 shows their grid search metrics. We chose $\gamma = 0.5$ and $\zeta = 1.0$ for TDS-FAPE, as it had the highest unique

success rate and the lowest average scRMSD and motif RMSD between the two motifs. For TDS-RMSD, several had high unique success rates, but we chose $\gamma = 2.0$ and $\zeta = 0.4$, which yielded the lowest scRMSD.

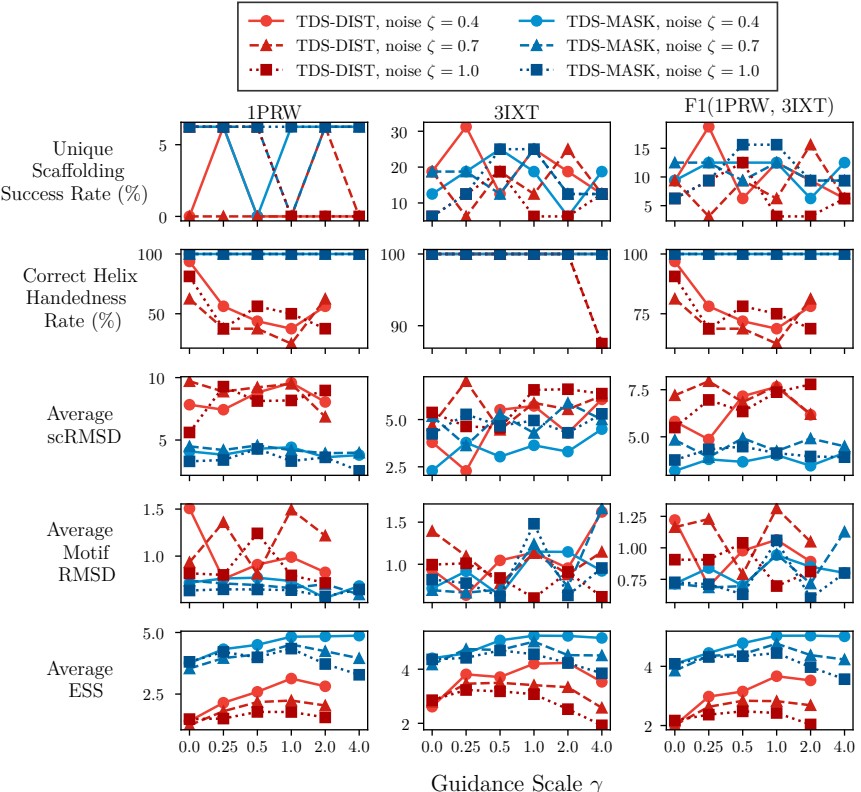

Figure 8: **Hyperparameter grid search for TDS-dist and TDS-mask**. The rates at which unique success and right-handed helices were achieved are reported. Average values for two of the four designability criteria are also shown. The right column depicts the F1 score between the values in the left and middle columns. TDS-DIST with $\gamma = 4.0$ is not shown in problem 1PRW, as it had numerical overflows in its backbone coordinates.

With TDS-FRAME-DIST, an additional parameter $\eta_{\mathrm{chiral}}$ is available to scale the chiral contribution of the rotation deviations to the likelihood. When $\eta_{\mathrm{chiral}} = 0$, we retrieve back TDS-DIST. Due to the large parameter space, we limit our search to a fixed temperature value of $\zeta = 0.4$ to match TDS-DIST. As shown in Fig. 10, a non-zero rotation scale indeed corrected for reflections from as little as $\eta_{\mathrm{chiral}} = 0.5$. However, as with 1PRW, there was a range of values between 0.5 and 4.0 where the rotation contribution was too weak, increasing the motif RMSD as it attempted to steer the trajectory away from solutions that meet the distance constraints to those with the correct handedness. When the scale was too large, solutions had the correct handedness but incorrect distances, leading to malformed backbones. Here, the scRMSD was at its highest. We found that the values $\eta_{\mathrm{chiral}} = 32.0$ and $\gamma = 0.25$ balance this trade-off but remark that the optimal value differed in the case of both motifs. For example, zero chiral contribution was necessary for motif problem 3IXT.

In summary, however, we acknowledge that our approach to hyperparameter tuning is limited. In particular, optimising for two select motifs is not guaranteed to generalise to other motif problems with different sizes, contiguity, and frequency in the data distribution. Where we can, we highlight possible reasons for our observations to help make more informed choices in the future. Furthermore, the computational expense of the procedure could not permit larger sample sizes in tuning nor a separate search for hyperparameters in the multi-motif case. Our results can certainly be improved, and the demonstrated albeit limited success in select multi-motif problems points to the feasibility of zero-shot approaches.

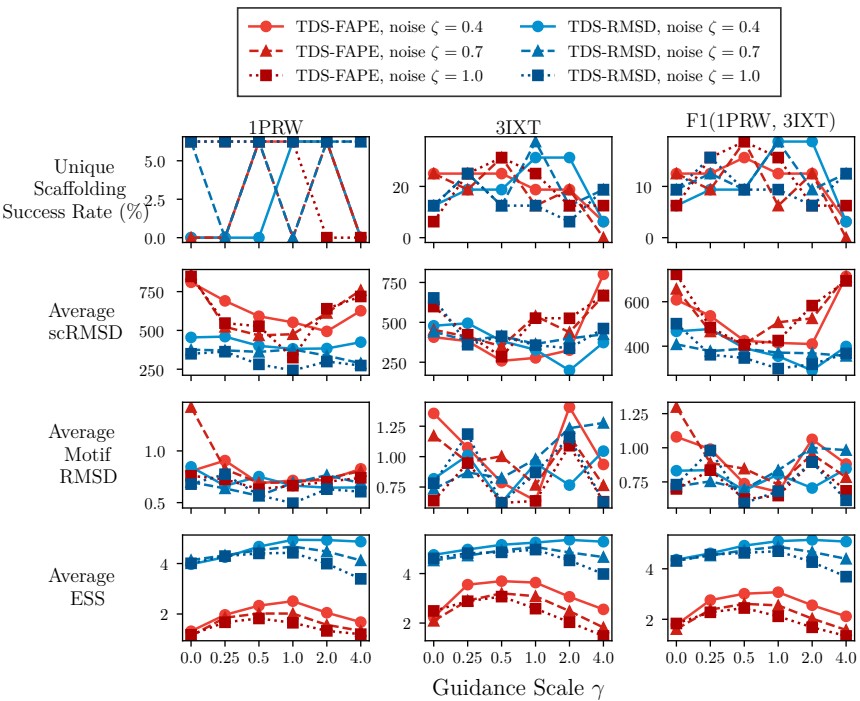

Figure 9: **Hyperparameter grid search for TDS-fape and TDS-rmsd**. The unique success rate and average values for two of the four designability criteria are shown. The right column depicts the F1 score between the values in the left and middle columns.

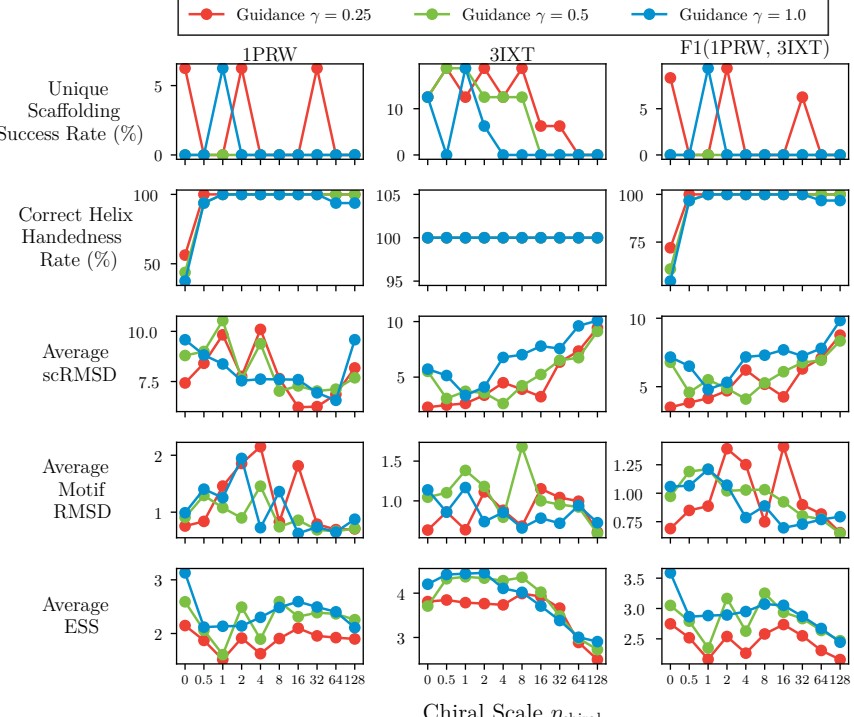

Figure 10: **Effect of chiral scale $\eta_{\mathrm{chiral}}$ on the handedness and designability of generated structures by TDS-frame-dist.** The right column depicts the F1 score between the values in the left and middle columns. A noise scale value of $\zeta = 0.4$ is used throughout.

## D.2 Exclusion of Sequence Requirements

The motif scaffolding problem, in full, requires both the motif's structure **m** and sequence **s** to be present in the designed proteins. The self-consistency pipeline described in Figure 2 precisely evaluates samples in this regard. The scRMSD measures the extent to which the generated structures can be produced by folding a protein sequence that contains **s** as a subsequence. In contrast, the motif RMSD measures the existence of **m** as a substructure of the generated structures.

However, zero-shot setups like ours only condition on the (backbone) structural information of the motif. Without providing sequence information to our posterior samplers, we effectively sample from the distribution of protein backbones that contain **m** as a substructure, despite them possibly having sequences that do not contain **s**. To quantify the success of our samplers at the structural aspect of the motif scaffolding task, we remove the fixing of the motif sequence in the self-consistency pipeline, allowing alternate sequences to be considered in place of the motif sequence. Fig. 11 shows that several single-motif problems that our samplers previously could not solve can be successfully scaffolded without the sequence requirements. As for problems that had a decrease in success, particularly in motif RMSD, we suspect this is due to a combination of stochasticity in the evaluation procedure and the loss of stability of the motif substructure that could have been achieved by fixing the sequence.

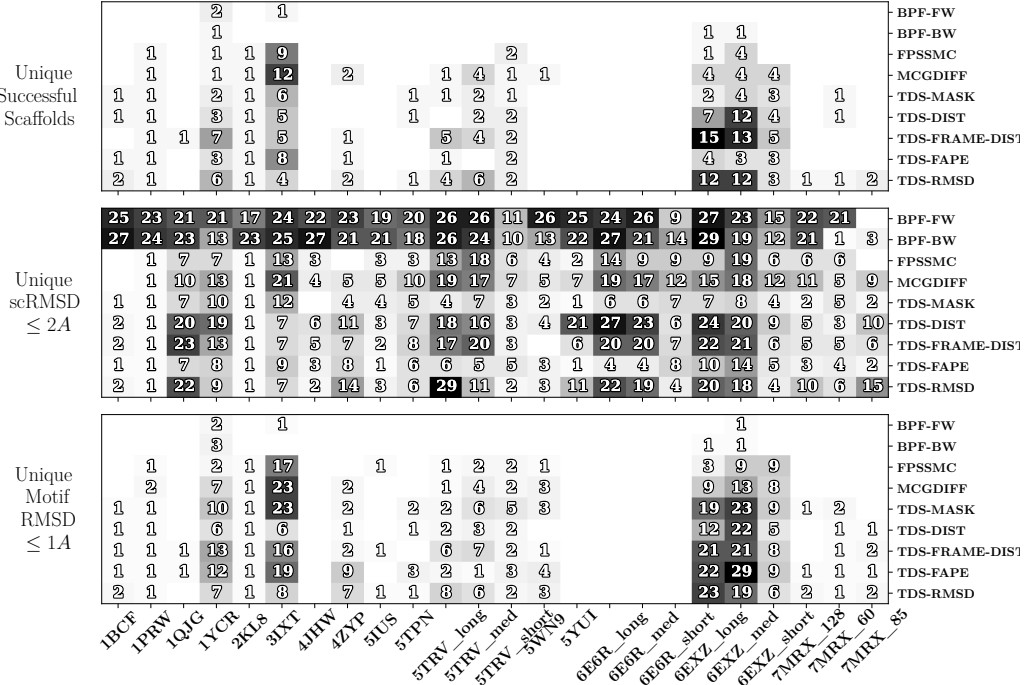

Figure 11: **Performance of sampling methods on the 24 motif scaffolding benchmarks when motif sequence is not fixed in the self-consistency pipeline.** Thirty-two backbones are sampled from each method across all the motif problems. Scaffolds that are successful and those which meet at least one of the main success criteria are reported according to their unique count.

Similarly, Fig. 12 shows the results for the multi-motif problems where the sequence requirements were omitted in the evaluation. TDS-RMSD solved the most problems under this looser success criteria. The difficulty with scaffolding multiple motifs is that the issues with sequence information are presumably compounded.

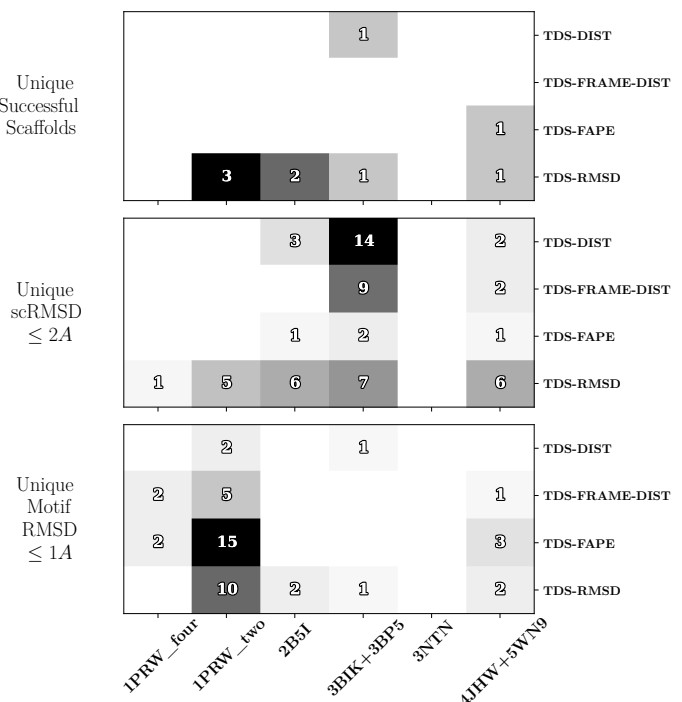

Figure 12: **Performance of sampling methods on the six multi-motif scaffolding benchmarks when motif sequences are not fixed in the self-consistency pipeline.** Thirty-two backbones are sampled from each method across all the motif problems. Scaffolds that are successful and those which meet at least one of the main success criteria are reported according to their unique count.

