# OpenReview forum: "On diffusion posterior sampling via sequential Monte Carlo for zero-shot scaffolding of protein motifs"
_TMLR — Accepted by TMLR_

### Review · Reviewer_CqDk · 2025-05-06

**Summary Of Contributions:**

This paper introduces a zero-shot approach to protein motif scaffolding using diffusion posterior sampling combined with Sequential Monte Carlo (SMC). The authors propose several new SE(3)-invariant guidance and frame-distance. The authors have conducted extensive experiments benchmarking different SMC-based samplers, showing that reconstruction-guidance methods with the proposed potentials match or outperform standard masking.
The major contribution of this work is generalizing posterior sampling to complex scaffolding tasks.

**Audience:**

Yes

**Claims And Evidence:**

Yes

**Requested Changes:**

1. It might be important to add variable motif placement or placement sampling to improve robustness of your results.
2. I am curious to see the computational costs for all these methods. Consider reporting them in the next version.
3. The details of how potential parameters are tuned are missing. It is advised to provide instructions on how those are selected.

**Strengths And Weaknesses:**

Strength:
1. First, introduce SE(3)-invariant potentials to generalize beyond masking, enabling symmetric scaffolding.
2. Extensive evaluations across 30+ tasks with multiple SMC-based samplers.

Weakness:
1. Evaluation scaling is quite limited. It seems to be restricted only to fixed motif placements and small sample sizes.
2. Can the authors comment on the multi-motif experimental results? It does not look too impressive to me.
3. One major drawback to me is the lack of essential coverage of the literature. Several Sequential Monte Carlo (SMC) variants are not discussed [1-2]. Compared to the more standardized SMC methods, many variants are reported to have less computational costs and can have even better performances in terms of reward values. These variants can probably also work with your novel potential, and it can be interesting to see it working. The authors could consider referring to [3] for a complete review of relevant inference-time techniques (i.e., SMC methods and variants).

[1] https://arxiv.org/pdf/2402.06320 [2] https://arxiv.org/pdf/2408.08252 [3] https://arxiv.org/abs/2501.09685

---

> ### Author Response · Authors · 2025-08-04
> **Replies to Reviewer CqDk**
>
> We thank the reviewer for their thoughtful review. Below, we respond to their main criticisms and the weaknesses they have pointed out.
>
> ### Weaknesses
>
> > Evaluation scaling is quite limited. It seems to be restricted only to fixed motif placements and small sample sizes.
>
> We certainly agree that our evaluation scaling may be limited, with fixed motif placements and only 16/32 replicates for each measurement in the single-motif and multi-motif problems, respectively. However, due to the combinatorial nature of our experiments, i.e. with 24 single-motif and six multi-motif problems, nine total methods considered, and 16 particles for each sampler instance, we are afraid the experimental setup in our paper was the only realistic configuration within our computational budget to evaluate the samplers. One of our main goals is to compare SMC samplers and guidance potentials -- hence, we focused on a fair comparison between these methods, rather than scaling up experiments to an industrial level.
>
>
> > Can the authors comment on the multi-motif experimental results? It does not look too impressive to me.
>
> Indeed, our results are incremental and more of a demonstration of what is possible rather than aiming to be state-of-the-art. Scaffolding multiple motifs *in a training-free manner* is a highly challenging problem, and we believe it requires a more principled way of tackling its associated inverse problem. By adapting SMC diffusion posterior samplers and an SE(3)-invariant guidance potential, we show it is, in fact, possible to solve it. While our results end here, we think this is certainly an exciting direction for future work to explore.
>
> On improving results: we echo our points in the paper that the main issues are due to (1) our motif placements are fixed, and (2) our inverse problem is defined structurally and does not incorporate the motif sequence. Fixed motif placements can be too restrictive, especially with multiple motifs, but were necessary to limit the variance of our comparisons between guidance potentials. We also found that, by not fixing the motif sequence in the evaluation, our setup was able to yield multi-motif successes, pointing to an underspecified target distribution.
>
> > One major drawback to me is the lack of essential coverage of the literature. Several Sequential Monte Carlo (SMC) variants are not discussed [1-2]. Compared to the more standardized SMC methods, many variants are reported to have less computational costs and can have even better performances in terms of reward values. These variants can probably also work with your novel potential, and it can be interesting to see it working.
>
> We thank the reviewer for pointing out these works. We have added these citations to our work in the background section (see the bottom of page 4 for a discussion of the references [1-2]).
>
> Regarding SVDD [1], we note that it can be viewed as multiple parallel instances of the bootstrap particle filter equipped with the DPS likelihood (eq. (4)). Its twisted version is recovered when TDS has its guidance scale set to zero. This case has already been considered in our hyperparameter search, where we found it to be sub-par to higher guidance scales.
>
> PDDS [2] specifically considers the case of sampling from a known target distribution (up to a normalising constant) which is unapplicable to our problem.
>
> ### Requested Changes
>
> > It might be important to add variable motif placement or placement sampling to improve robustness of your results.
>
> With multiple SMC samplers to evaluate, we focused on reducing the variance of our results and avoiding any unnecessary complexity to arrive at a fair comparison. Our scheme of fixing the motif placement follows directly from the setup of Wu et al. (TDS paper), which we believe is still representative of the design challenges faced in motif scaffolding problems.
>
> We would like to additionally note that scaffolding with degrees of freedom requires specifying the maximum number of placements to consider (given it combinatorially blows up for discontiguous single-motif or multi-motif problems) which can introduce additional complexity. We think dedicated methods for this problem could be explored in a work that considers this setting specifically.

---

> > ### Author Response · Authors · 2025-08-04
> > **Replies to Reviewer CqDk, continued**
> >
> > > I am curious to see the computational costs for all these methods. Consider reporting them in the next version.
> >
> > To address this, we have added a new results section, Section 4.5, that goes over the computational costs of our methods.
> >
> > Tables 1 and 2 (on page 14) of the updated manuscript summarise our findings. As expected, samplers relying on backward passes to the score model had slower runtimes. On the other hand, for certain methods, we found a simple optimisation can lead to significant speed-ups. By only evaluating the score model for *unique* particles, the cost becomes proportional to the effective sample size (ESS). While we could label these methods as pursuing a trade-off in speed, it is a result of weight degeneracy which is a sign of inefficiency.
> >
> > > The details of how potential parameters are tuned are missing. It is advised to provide instructions on how those are selected.
> >
> > Frame-distance was the only guidance potential to have a tuned parameter in its chiral scale. We have indeed done a hyperparameter search for TDS-FRAME-DIST (Fig. 10) - we have the chiral scale as one of the grid dimensions (i.e. it is searched jointly with the TDS parameters). We apologise if this was unclear.

---

### Review · Reviewer_a1ij · 2025-06-04

**Summary Of Contributions:**

At a highlevel this paper applies SMC based methodologies for diffusion such as TDS (Wu et al 2023) to a newly defined likelihood for motif scaffolding that combining different guidance potentials each of which incorporate better inductive biases / probabilistic modelling assumptions for the mottif scaffolding task.

The empirical results seem compelling and the proposed guidance potentials are novel and well motivated however the authors need to work harder to present the overall method.

**Audience:**

Yes

**Claims And Evidence:**

Yes

**Requested Changes:**

1. Please start by including an algorithm with pseudocode of your overall scheme in the main, well summarised, and possibly colour-coded to highlight differences between TDS and SMCDiff as much as possible (what adaptations were necessery to incorporate these new guidances).  I can see there are a lot  of algorithms in the Appendix but it has not been structured / layed out in a way that is easy to review.
2. Define all core terms of equations in the main such that the main equations are self-contained. I shouldn't have to go and read the Song inverse problem paper to figure out your notation. See weaknesses in particular if the filtering model implied by equation (2) is important to your method (eq 2 takes up a very central location in the paper ...) please define it in more detail and motivate why its relevant.

**Strengths And Weaknesses:**

Weaknesses

1. The paper is sometimes too informally written examples:
    a. "One idea is "
    b. "complicated algorithms" (what makes them complicated ???)
2. The notation lacks defining things, for example, in equation (2), what is p(y_t|y) ?

Typically when doing conditional genreative modelling and inverse problems we have:

$$
p(y|x_0) = \mathcal{N}(y| A(x_0) , \tau I)
$$

And the conditional score (not an approximation ) which is guaranteed (under certain assumptions on the noise distribution/noise schedule [1,2 ]) is given by :

$$
\nabla_{x_t} \ln p_{t}(x_t|y) = \nabla_{x_t} \ln  \int p(x_t, y, x_0) dx_0 =   \nabla_{x_t} \ln  \int p(y| x_t, x_0) p(x_t|x_0) dx_0
$$
Where $ p(y| x_t, x_0) =  p(y| x_0)$ is not an approximation, however, the decomposition of the conditional score you have written of the score is different, so this is not an issue, but just poiting out why what you wrote seemed confusing at first. I assume is applying the product rule of prob in a different direction, I dont think its wrong but its difficult to read with the level of provided detail. What is $y_t$ ?  you seemed to have defined a full state space model / filtering process in a single sentence :

```
One idea is to project the intermediate latent variables xt onto a measurement subspace at each denoising
```

This is just not enough detail.


[1] Denker, A., Vargas, F., Padhy, S., Didi, K., Mathis, S., Barbano, R., Dutordoir, V., Mathieu, E., Komorowska, U.J. and Lio, P., 2024. DEFT: Efficient Fine-tuning of Diffusion Models by Learning the Generalised $ h $-transform. Advances in Neural Information Processing Systems, 37, pp.19636-19682.

[2] Domingo-Enrich, C., Drozdzal, M., Karrer, B. and Chen, R.T., 2024. Adjoint matching: Fine-tuning flow and diffusion generative models with memoryless stochastic optimal control. arXiv preprint arXiv:2409.08861.

---

> ### Author Response · Authors · 2025-08-04
> **Replies to Reviewer a1ij**
>
> We thank the reviewer for their valuable and thoughtful feedback. We have done our best to follow their advice and believe this has substantially improved the clarity of our presentation on diffusion posterior sampling and SMC samplers. Below, we address their main points and requests.
>
> ### Weaknesses
>
> > The paper is sometimes too informally written examples: a. "One idea is " b. "complicated algorithms" (what makes them complicated ???)
>
> We thank the reviewer for pointing this out. Indeed, we have gone through the full text again and resolved these issues.
>
> > The notation lacks defining things, for example, in equation (2), what is p(y_t|y) ?
>
> We apologise for not being verbose enough. This notation, now denoted as $\nu(\mathbf{y}\_t \mid \mathbf{y}\_0)$, refers to the distribution of the ''noisy'' version of the measurement at time $t$, conditioned on the original (clean) measurement $\mathbf{y}\_0 = \mathbf{y}$. We present it generally in our background text, but, as far as we know, it is only tractable for linear inverse problems. We discuss how the linear structure allows one to define the noisy measurement process $\mathbf{y}\_t$ in App. A.2. (eq. 11).
>
> > Typically when doing conditional genreative modelling and inverse problems we have: $$p(y|x_0) = \mathcal{N}(y| A(x_0) , \tau I)$$ And the conditional score (not an approximation) [...] the decomposition of the conditional score you have written of the score is different, so this is not an issue, but just poiting out why what you wrote seemed confusing at first. I assume is applying the product rule of prob in a different direction, I dont think its wrong but its difficult to read with the level of provided detail.
>
> We thank the reviewer for pointing this out. This was indeed a misprint (which read "approximation" in the old version of our manuscript). We have now corrected this. We use the following formula to decompose the score of the distribution $p(\mathbf{x}_t \mid \mathbf{y})$, i.e. we have from the Bayes' formula (now stated in eq. (2) in our paper)
>
>
> $$\nabla\_{\mathbf{x}\_{t}}{\log q(\mathbf{x}\_{t} \mid \mathbf{y})} = \nabla\_{\mathbf{x}\_t} \log q(\mathbf{x}\_t) + \nabla\_{\mathbf{x}\_{t}}{\log g(\mathbf{y} \mid \mathbf{x}\_{t})},$$
>
>
> where $\nabla\_{\mathbf{x}\_t} \log q(\mathbf{x}\_t)$ is the unconditional (pretrained) score and $\nabla\_{\mathbf{x}\_{t}}{\log g(\mathbf{y} \mid \mathbf{x}\_{t})}$ is the guidance term. This is a simple consequence of the Bayes' rule and used in [1-3]. We hope that in our updated Section 2 this has become clear, please let us know if not.
>
>
> > What is $y_t$? you seemed to have defined a full state space model / filtering process in a single sentence :
> >
> >>One idea is to project the intermediate latent variables xt onto a measurement subspace at each denoising
> >
> >This is just not enough detail.
>
> We thank the reviewer for pointing this out. We have comprehensively rephrased this part in our work to clarify how Song. et al (2020) built their construction.
>
> ### Requested changes
>
> > Please start by including an algorithm with pseudocode of your overall scheme in the main, well summarised, and possibly colour-coded to highlight differences between TDS and SMCDiff as much as possible (what adaptations were necessery to incorporate these new guidances). I can see there are a lot of algorithms in the Appendix but it has not been structured / layed out in a way that is easy to review.
>
> To follow this, we have added a unified and general recipe for SMC diffusion posterior samplers (Alg. 1) in the background text, including some explanations of techniques used by the samplers, e.g. twisting targets and tilting the proposal. We have restructured the appendix (now App. A.2) to precisely define the design choices of all samplers in a consistent notation, a summary of which is available in Table 3 (on page 21). As we noticed the weight computations may be a source of confusion, we decided to add their derivations. How the samplers were adapted for the guidance potentials are laid out in App. C.2.

---

> > ### Author Response · Authors · 2025-08-04
> > **Replies to Reviewer a1ij, continued**
> >
> > > Define all core terms of equations in the main such that the main equations are self-contained. I shouldn't have to go and read the Song inverse problem paper to figure out your notation. See weaknesses in particular if the filtering model implied by equation (2) is important to your method (eq 2 takes up a very central location in the paper …) please define it in more detail and motivate why its relevant.
> >
> > We apologise for not being precise in the background and having left out definitions for some terms. We have made changes to be more explicit in our writing. Notably, we expand more on the noisy measurement process $\mathbf{y}_t$ (and its construction for linear problems in App. A.2). We also pin the intractability of comparing a clean measurement with a noisy latent as a clear motivation for the approach taken by the two approximations (noising the measurement or denoising the latent). Please let us know if there are still areas we should provide more detail on.
> >
> > **References**
> >
> > [1] Chung, Hyungjin, et al. "Diffusion Posterior Sampling for General Noisy Inverse Problems." International Conference on Learning Representations (ICLR). 2023.
> >
> > [2] Song, Jiaming, et al. "Pseudoinverse-guided diffusion models for inverse problems." International Conference on Learning Representations (ICLR). 2023.
> >
> > [3] Boys, Benjamin, et al. "Tweedie Moment Projected Diffusions for Inverse Problems." Transactions on Machine Learning Research. 2024.

---

### Review · Reviewer_6Enk · 2025-07-23

**Summary Of Contributions:**

This paper proposes a method for protein scaffold design using pretrained general models $q(x)$ while enforcing specific motifs. This is done by fixing the motifs and using diffusion guidance + SMC to solve the conditional posterior sampling problem $\propto q(x)g(y|x)$. Specifically, the authors argue that prior work defines $g$ in counterproductive ways because they work directly in coordinate space, $x$, and its gradients perturb the motifs that we are trying to keep fixed thus creating a suboptimal guidance path when masking; instead, they propose to define $g$ in internal coordinates/in an SE(3)-invariant way so that the diffusion guidance keeps the motif intact.

This new guidance is defined by a combination of RMSD, frame-aligned point error (as in AF2), and internal pairwise distances.

The paper then shows how this enables sampling, as well as doing more complex motif conditioning, in fairly hard settings.

**Audience:**

Yes

**Claims And Evidence:**

Yes

**Requested Changes:**

I think the paper is good as is, more investigation-type results as I've alluded to above could help but are not strictly necessary.

Some other suggestions:
- There's no clear analysis of the computational costs of the methods involved here. Are there trade-offs?
- It seems like there is high hyperparameter sensitivity; while that analysis is welcome, it would be good to understand why, or perhaps to suggest a guide to the reader.
- If possible, improving the multi-motif experiments with more evidence points.

**Strengths And Weaknesses:**

The paper is well written and tackles a fairly complex problem with elegance. The empirical results are generally enough to back up the paper's claims, although in the multi-motif experiments, the results are more "signs of life" than strong quantitative results. Generally, the paper does offer some insights into why their method works, but these insights are in a way "negative" (e.g. handedness results); they show that a series of steps are needed to get SE(3) guidance right and they point out some challenges of the problem, but there is no very deep comparison with prior work.

In terms of TMLR's second criteria, audience, I think the paper is very timely and will find many readers.

---

> ### Author Response · Authors · 2025-08-04
> **Replies to Reviewer 6Enk**
>
> We are pleased the reviewer has found the paper timely and its presentation of the problem accessible. We also thank them for their feedback. Below, we provide some comments to their points:
>
> ### Weaknesses
>
> > in the multi-motif experiments, the results are more "signs of life" than strong quantitative results
>
> We agree that our multi-motif results are more of a demonstration of what is possible rather than aiming to be state-of-the-art, and we have made some revisions to be more transparent of the fact. Echoing our reply to another reviewer who raised the same point, scaffolding multiple motifs *in a training-free manner* is a highly challenging problem, we believe it requires a more principled way of tackling its associated inverse problem. By adapting SMC diffusion posterior samplers and an SE(3)-invariant guidance potential, we show it is, in fact, possible to solve it. While our results end here, we think this is certainly an exciting direction for future work to explore.
>
> > there is no very deep comparison with prior work
>
> We wanted to limit our analyses to SMC diffusion posterior samplers, seeing as they were the most performant training-free methods available. Among which, only SMCDiff and TDS had been evaluated on motif scaffolding. Because there are also several moving parts when tackling the problem, e.g. the base diffusion model, success criteria, and motif specifications, we chose to compare existing works in a controlled setting to ensure a fair comparison.
>
> ### Requested Changes
>
> > There's no clear analysis of the computational costs of the methods involved here. Are there trade-offs?
>
> To address this, we have added a new results section, Section 4.5, that goes over the computational costs of our methods.
>
> Tables 1 and 2 (on page 14) of the updated manuscript summarise our findings. As expected, samplers relying on backward passes to the score model had slower runtimes. On the other hand, for certain methods, we found a simple optimisation can lead to significant speed-ups. By only evaluating the score model for *unique* particles, the cost becomes proportional to the effective sample size (ESS). While we could label these methods as pursuing a trade-off in speed, it is a result of weight degeneracy which is a sign of inefficiency.
>
> > It seems like there is high hyperparameter sensitivity; while that analysis is welcome, it would be good to understand why, or perhaps to suggest a guide to the reader.
>
> We thank the reviewer for pointing this out. There are two things we can speak about this.
>
> First are the motif problems we tune our hyperparameters against. As it is too expensive to run a grid search on the entire motif benchmark, a subset of motifs, representative of different design challenges, is ideal to get a well-rounded summary of sampler performance. We chose two motifs, one contiguous (3IXT) and one discontiguous (1PRW), for our experiments, without a strong reason to go for either one. We suggest future works to approach this motif subset curation more principledly according to, say, the number of segments, type of fold, and perhaps even their biological significance. We also believe motif "complexity" is one reason for the hyperparameter sensitivity, e.g. a higher guidance scale is required to push backbone atoms to adopt a rare or unusual fold but it may "overshoot" for simpler motifs where the unconditional score is sufficient.
>
> Second are the many interacting parameters in the sampler alone. For example, our base diffusion model, Genie, has a hyperparameter in its noise/temperature scale. While it is not entirely principled, it is supposed to draw parallels to annealing its support distribution, where a trade-off between higher diversity and higher quality samples can be made. And, in fact, they find it provides a substantial improvement in their unconditional generation. How this interacts with conditional generation is non-trivial and has to be verified empirically. One might want to prioritise higher quality samples to maximise the success rate for difficult motif problems or if they have a limited compute budget, and another might prefer a higher diversity if the previous samples were nearly identical.
>
> > If possible, improving the multi-motif experiments with more evidence points.
>
> We now more explicitly refer to our findings in App. D.2, where not fixing the motif sequence in the evaluation led to successes in the original 32-replicate multi-motif experiments, as evidence that the samplers successfully target backbones with the desired structure but further need to incorporate sequence information in the inverse problem statement to satisfy the requirements of the self-consistency pipeline.

---

### Author Response · Authors · 2025-08-04
**Summary of Changes**

We sincerely thank the reviewers for their constructive feedback. Incorporating the suggestions we received, we have made the following main revisions to the paper (with changes highlighted in **blue**):
- We added a general recipe unifying the SMC diffusion posterior samplers in the background text (Alg. 1), with an accompanying breakdown summary of all the samplers' specific design choices in App. A.2.
- We report the computational costs of samplers and guidance potentials in the results (Section 4.5). We also present empirical runtime comparisons between samplers.
- We draw more attention to the significance of measurement-tilted proposals and twisted targets. We give brief explanations in the background and some intuition on how these techniques improved upon the BPF results.
- We made several changes to the main text to improve the overall clarity, readability, and flow of our ideas.

---

### Decision · Action_Editor_9LpX · 2025-09-08

**Recommendation:** Accept as is

**Additional Comments:**

None.

**Audience:**

Yes

**Audience Explanation:**

All reviewers agree that this work is interesting for the ML community as well as for the bio community.

**Claims And Evidence:**

Yes

**Claims Explanation:**

All the reviewers agree with that this work present value for the ML community. The technical contributions are sound and the authors have successfully addressed the comments of the reviewers. This work represents an interesting contribution to investigation of posterior sampling with diffusion models.The experiments are run rigorously with clear success criteria.